

# Probabilistic projections of the Amery Ice Shelf catchment, Antarctica, under high ice-shelf basal melt conditions

Sanket Jantre[1], Matthew J. Hoffman[2], Nathan M. Urban[1], Trevor Hillebrand[2], Mauro Perego[3], Stephen Price[2], and John D. Jakeman[3]

[1]Applied Mathematics Group, Computational Science Initiative, Brookhaven National Laboratory, Upton, NY, USA
[2]Fluid Dynamics and Solid Mechanics Group, Los Alamos National Laboratory, Los Alamos, NM, USA
[3]Center for Computing Research, Sandia National Laboratories, Albuquerque, NM, USA

**Correspondence:** Sanket Jantre (sjantre@bnl.gov)

**Abstract.** Antarctica's Lambert Glacier drains about one-sixth of the ice from the East Antarctica Ice Sheet and is considered stable due to the strong buttressing provided by the Amery Ice Shelf. While previous projections of the sea-level contribution from this sector of the ice sheet have predicted significant mass loss only with near complete removal of the ice shelf, the ocean warming necessary for this was deemed unlikely. Recent climate projections through 2300 indicate that sufficient ocean warming is a distinct possibility after 2100. This work explores the impact of parametric uncertainty on projections of the Lambert-Amery system's (hereafter "Amery sector") response to abrupt ocean warming through Bayesian calibration of a perturbed-parameter ice-sheet model ensemble. We address the computational cost of uncertainty quantification for ice-sheet model projections via statistical emulation, which employs surrogate models for fast and inexpensive parameter space exploration while retaining critical features of the high-fidelity simulations. To this end, we build Gaussian process (GP) emulators from simulations of the Amery sector at medium resolution (4-20 km mesh) using the MPAS-Albany Land Ice (MALI) model. We consider six input parameters that control basal friction, ice stiffness, calving, and ice-shelf basal melting. From these, we generate 200 perturbed input parameter initializations using space-filling Sobol sampling. For our end-to-end probabilistic modeling workflow, we first train emulators on the simulation ensemble then calibrate the input parameters using observations of the mass balance, grounding line movement, and calving front movement with priors assigned via expert knowledge. Next, we use MALI to project a subset of simulations to 2300 using ocean and atmosphere forcings from a climate model for both low and high greenhouse gas emissions scenarios. From these simulation outputs, we build multivariate emulators by combining GP regression with principal component dimension reduction to emulate multivariate sea-level contribution time series data from the MALI simulations. We then use these emulators to propagate uncertainty from model input parameters to predictions of glacier mass loss to 2300, demonstrating that the calibrated posterior distributions have both greater mass loss and reduced variance than the uncalibrated prior distributions. Parametric uncertainty is large enough through about 2130 that the two projections under different emissions scenarios are indistinguishable from one another. However, after rapid ocean warming in the first half of the twenty-second century, the projections become statistically distinct within decades. Overall, this study demonstrates an efficient Bayesian calibration and uncertainty propagation workflow for ice-sheet model projections and identifies the potential for large sea-level rise contributions from the Amery sector of the Antarctic Ice Sheet after 2100 under high greenhouse gas emission scenarios.



# 1 Introduction

With an area of slightly more than 60,000 km$^2$ (Andreasen et al., 2023), the Amery Ice Shelf (AmIS) is the third largest
ice shelf in Antarctica and drains approximately 16% of the ice from East Antarctica (Fricker et al., 2002) (Fig. 1). AmIS is
considered particularly stable due to its location in a narrow embayment with many pinning points (Pittard et al., 2017); the
convergent flow of Lambert, Fisher, and Mellor Glaciers entering the ice shelf (Gong et al., 2014); and a prograde bed slope
beneath the grounded ice feeding the ice shelf (Morlighem et al., 2020). The most recent in-depth model projection study of
the Amery sector (Lambert-Amery system) of the Antarctic Ice Sheet predicts an insignificant sea-level contribution from this
sector for the next 500 years, barring extreme ocean temperature increases, which at that time was considered unlikely (Pittard
et al., 2017).

However, new projections of the Antarctic Ice Sheet using global climate model (GCM) ocean and atmospheric conditions
through 2300 indicate that extreme ocean warming and subsequent ocean- and/or surface melt-driven removal of most major
ice shelves, including the AmIS, is possible after 2100 (Seroussi et al., 2023). Removal or extensive thinning of the ice shelf
will accelerate the grounded ice through a reduction in buttressing (Gudmundsson et al., 2019; Zhang et al., 2020), causing
a substantially larger contribution to sea-level rise than previously projected. Ice-shelf cavities in Antarctica generally can
be characterized as "cold" or "warm," depending on whether their circulation and melting are controlled by cold and saline
Shelf Water or deeper and relatively more warm saline-modified Circumpolar Deep Water (mCDW) (Dinniman et al., 2016).
Changes in the access of these water masses to the ice-shelf base can lead to a regime shift and a rapid change in ice-shelf
basal melt rates of an order of magnitude (Hazel and Stewart, 2020; Haid et al., 2023). Under present conditions, only small
intrusions of mCDW reach the AmIS (Liu et al., 2017). However, in a high greenhouse gas emissions scenario, more pervasive
access of mCDW to the ice-shelf base appears possible after 2100 as shown in the four GCM projections used by Seroussi and
Nowicki (2024) (Fig. 2). With these projections indicating the potential for removal of the AmIS through increased submarine
melting, the long-term stability of the AmIS sector should be reevaluated.

Ice-sheet models inherently rely on uncertain model parameterizations (e.g., for representing unresolved sub-grid-scale
physical processes or unobserved ice mechanical and thermodynamic characteristics) that introduce uncertainty into their pre-
dictions of a glacier's dynamic response to changes in climate forcing. Therefore, assessing the AmIS sector's response to
a potential sudden increase in ice-shelf basal melting during the twenty-second century requires thorough quantification of
this parametric uncertainty. Quantifying how uncertainties in input parameters influence future predictions is often achieved
through uncertainty propagation, typically using Monte Carlo sampling over the input space. Bayesian calibration, or proba-
bilistic parameter estimation, extends uncertainty propagation by providing a systematic framework for integrating observa-
tional data and expert knowledge to constrain the distribution of input uncertainties. This, in turn, allows for the generation of
observationally constrained probabilistic projections. Bayesian calibration not only enhances the reliability of ice-sheet model
projections but also provides insights into the sources and impacts of uncertainties on those projections. This calibration work-
flow, which typically requires Monte Carlo sampling over computationally expensive parametric model ensembles, becomes
tractable when combined with statistical emulation of the expensive ice-sheet model. The result is a rigorous method for quan-





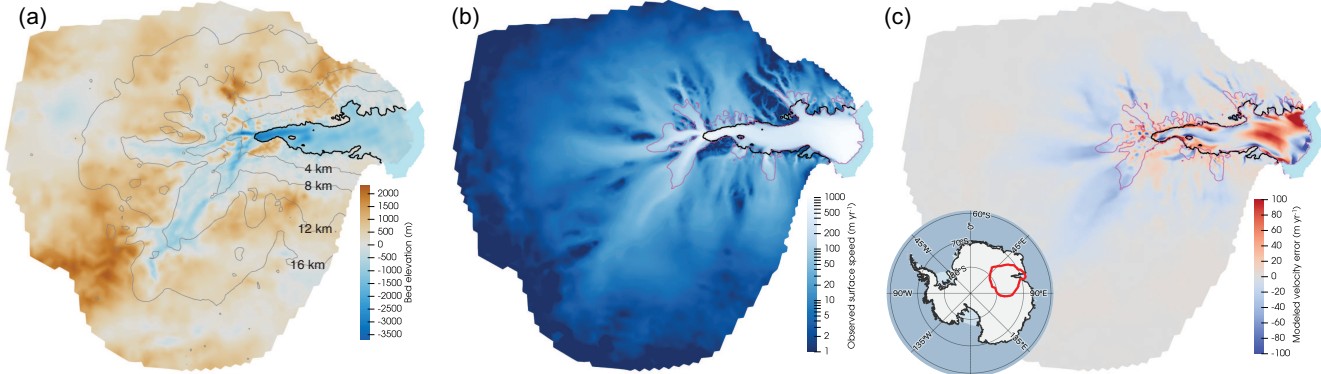

**Figure 1.** Model domain encompassing the Amery Ice Shelf and the Lambert Glacier catchment. Red line in panel c inset map shows the catchment location within the broader Antarctica Ice Sheet. a) Bedrock elevation (color) with gray lines contouring MALI mesh resolution. b) Observed ice surface speed. c) Difference between modeled, optimized initial, and observed ice surface speeds. In all panels, the black line represents the grounding line, and light blue fill indicates open ocean in the model domain. The pink line in panels b and c is the observed 50 m yr$^{-1}$ ice surface speed contour.

tifying and communicating the reliability of model predictions, which aids decision-making in climate science and policy. Numerous studies have introduced formal uncertainty quantification methods into ice-sheet modeling to generate probabilistic projections of future contributions to sea-level change from various glaciers and ice sheets (Bulthuis et al., 2019; Nias et al., 2019; Alevropoulos-Borrill et al., 2020; Gilford et al., 2020; Wernecke et al., 2020; Lee et al., 2020; Edwards et al., 2021; Berdahl et al., 2021; Hill et al., 2021; Aschwanden and Brinkerhoff, 2022; Chang et al., 2022; Berdahl et al., 2023; Bevan

et al., 2023; Johnson et al., 2023; Nias et al., 2023; Seroussi et al., 2023; Van Katwyk et al., 2023). In this study, we apply such methods to the AmIS catchment for the first time to answer the following science question:

*How does parametric uncertainty, when constrained by observations, affect the projected ice-sheet response to abrupt change in oceanic forcing of the Amery Ice Shelf sector of Antarctica?*

To address this question, we apply a regional ice-sheet model of the AmIS catchment to generate an ensemble of simulations

with parameter values perturbed over their likely ranges. Using statistical emulation of the resulting ensemble, we employ Bayesian calibration of the uncertain model parameters, using observations of key glacier quantities during the historical period. By sampling from the posterior distribution of parameter values and combining ensembles of ice-sheet model projections with statistical emulation, we generate calibrated, probabilistic projections of the future contribution of the AmIS sector to sea level under two climate scenarios.

In this work, we first present the ice-sheet model configurations used in our study, followed by a detailed description of the Bayesian modeling framework, including the statistical emulation that enables efficient Monte Carlo sampling. Finally, we share our results and discuss their implications for the future of the AmIS sector of the East Antarctic Ice Sheet (EAIS). We also discuss how these methods may be extended to more complex problems, including the entire Antarctic Ice Sheet.





## 2 Ice-sheet model description

For our simulations, we use the MPAS-Albany Land Ice (MALI) model (Hoffman et al., 2018) applied to a regional domain of the AmIS catchment (Fig. 1). Here, we describe the main model features employed in this study, highlighting equations with uncertain model parameters, and refer the reader to references for other details.

### 2.1 Model configuration

MALI is a variable resolution mesh ice-sheet model that solves the first-order, three-dimensional (3D), Blatter-Pattyn approx-
imation to the Stokes equations for momentum balance using the finite element method. We use the common constitutive relation $\tau_{ij} = 2\eta_e \dot{\epsilon}_{ij}$ , where $\tau_{ij}$ is the deviatoric stress tensor, $\dot{\epsilon}_{ij}$ is the strain rate tensor, and $\eta_e$ is the effective ice viscosity given by Nye's generalization of Glen's flow law (Glen, 1955; Nye, 1957):

$$\eta_e = C_\phi \phi A^{-\frac{1}{n}} \dot{\epsilon}_e^{\frac{1-n}{n}}, \tag{1}$$

where $A$ is a temperature-dependent rate factor, $n$ is an exponent taken as 3 for polycrystalline glacier ice, $\phi$ is a spatially
varying ice stiffness factor accounting for the impacts of unresolved processes (e.g., fabric) on ice rheology, and $C_\phi$ is a spatially uniform ice-stiffness adjustment factor taken as an uncertain parameter. We use a power law for basal friction of the form

$$\tau_b = C_\mu \mu |u_b|^{q-1} u_b, \tag{2}$$

where $\tau_b$ is basal shear stress, $u_b$ is the slip velocity at the glacier bed, and $0 < q \leq \frac{1}{3}$ is an uncertain power law exponent
representing the degree of plasticity of the bed. $\mu$ is a spatially varying friction parameter, and $C_\mu$ is a scalar basal friction adjustment factor taken as an uncertain parameter.

Thickness and tracer (temperature) advection is performed with a first-order upwind scheme implemented with the finite volume method. The configuration here employs thermomechanical coupling and a temperature-based thermal solver. MALI uses a forward Euler time integration scheme with an adaptive time step that here is selected to be 0.2 of the time interval
defined by the advective Courant–Friedrichs–Lewy (CFL) condition. This is smaller than what is typically chosen for MALI, but it minimizes the chances of unstable model behavior when ignoring the diffusive CFL condition and exploring a wide range of parameter space.

We also employ parameterizations for calving, ice-shelf basal melting, and submarine melting of grounded marine termini (as described in Hillebrand et al. (2022)). MALI's subglacial hydrology model (Hager et al., 2022) and glacier isostatic adjustment
(Book et al., 2022) are not used in this study.

For calving, we apply the von Mises stress calving parameterization from Morlighem et al. (2016),

$$c = |\overline{u}| \frac{\sigma}{\sigma_{max}}, \tag{3}$$

where $c$ is calving velocity, $\overline{u}$ is depth-averaged ice velocity, $\sigma$ is depth-averaged tensile von Mises stress, and $\sigma_{max}$ is a yield stress treated as a tuning parameter. Because there are negligible grounded marine calving fronts currently in the Amery





catchment, it is not possible to calibrate grounded margin calving, which, in practice, may require a different yield stress parameter than that for floating ice (Choi et al., 2017; Hillebrand et al., 2022). Furthermore, Hillebrand et al. (2022) identifies a possible positive feedback between grounded calving and basal slip when both $\sigma_{max}$ and $q$ are small, leading to unrealistically catastrophic glacier retreat – behavior that echoed in preliminary runs of this work. To avoid these complexities given the large range of parameters being considered, we disable grounded calving entirely. Thus, the glacier retreat in simulations where

significant grounded calving fronts develop (following loss of ice shelves) should be considered a conservative projection as it is likely underestimated under such conditions.

For ice-shelf basal melting, we use the scheme by Jourdain et al. (2020), which is designed for Antarctic ice shelves and prescribed in the ISMIP6-AIS experimental protocol (Nowicki et al., 2020; Seroussi et al., 2020). This parameterization defines spatially varying ice-shelf basal melt rates, $m$, as a function of ocean thermal forcing along the ice-shelf base, $TF$, the

difference between the ocean temperature and the local ocean freezing temperature:

$$ m = \gamma_0 \left( \frac{\rho_{sw} c_{pw}}{\rho_i L_f} \right)^2 (TF + \delta T) \left| \langle TF \rangle + \delta T \right|, \tag{4} $$

where $\rho_{sw}$ is the density of seawater, $c_{pw}$ is the specific heat of seawater, $\rho_i$ is the density of glacier ice, $L_f$ is the latent heat of fusion of ice, and $\langle TF \rangle$ is the thermal forcing averaged over the entire ice-shelf base. The coefficient $\gamma_0$ is an uncertain proportionality constant, and $\delta T$ is an uncertain bias correction factor.

For grounded marine termini, the melt rate perpendicular to the horizontal calving front, $m_g$, is parameterized using the form $m_g = (A h q_{sq}^{\alpha} + B) TF_b^{\beta}$ (Rignot et al., 2016; Slater et al., 2020), where $A = 0.0003 \ \text{m}^{\alpha} \ \text{d}^{\alpha-1} \ {}^{\circ}\text{C}^{-\beta}$, $h$ is water depth at the terminus in meters, $q_{sq}$ is subglacial runoff in meters per day, $\alpha = 0.39$, $B = 0.15 \ \text{m d}^{-1} \ {}^{\circ}\text{C}^{-\beta}$, $\beta = 1.18$, and $TF_b$ is the ocean thermal forcing at the bed depth. Because the Amery catchment currently has negligible grounded marine margins, we are unable to tune this process and instead use the standard parameter values prescribed for ISMIP6-Greenland (Slater et al.,

2020) without uncertainty and conservatively assume $q_{sq} = 0$. Despite being negligible in the initial state, significant grounded marine termini develop late in some future scenarios after ice shelves have largely disappeared.

## 2.2 Amery Ice Shelf catchment domain

The regional simulation domain is defined by the Amery B-C region used in Rignot et al. (2019), which encompasses all ice flowing into the AmIS (Fig. 1). This regional domain is extracted from the whole Antarctic domain used by MALI for the

ISMIP6-Projections2300-Antarctica intercomparison (Seroussi et al., 2023). The mesh resolution is 4 km in areas of fast flow ($\log_{10}(u_s) > 2.5$, where $u_s$ is observed ice surface speed in m yr$^{-1}$) or near the 2015 grounding line ($< 10$ km) and coarsens to 20 km in locations of slow flow ($\log_{10}(u_s) < 0.75$) or far (>100 km) from the grounding line (Fig. 1(a)). The mesh contains 53,523 total cells. The vertical coordinate has five terrain-following layers with higher resolution near the bed. The overall moderate mesh resolution is chosen to make a large ensemble of simulations computationally feasible.

Ice thickness and bed topography are interpolated from BedMachine Antarctica v2 (Morlighem et al., 2020; Morlighem, 2022) using conservative remapping. The spatially varying and time invariant basal friction ($\mu$) and ice stiffness ($\phi$) fields are solved for the entire Antarctic Ice Sheet using a partial-differential-equation-constrained optimization problem (Perego



et al., 2014), minimizing the misfit of ice surface velocity relative to observations from 1996 to 2016 (Mouginot et al., 2017)
(Fig. 1(c)). The solution to the optimization problem satisfies the momentum balance and steady-state thermodynamics, yield-
ing a consistent initial ice temperature and velocity field. For both optimization and forward simulations, the thermal basal
boundary condition is provided by the geothermal flux map of Shapiro and Ritzwoller (2004), and the surface thermal bound-
ary condition is the mean annual air temperature from the RACMO2.1 1979–2010 climatology (Lenaerts et al., 2012).

## 2.3 Climate scenarios

Four climate forcing scenarios are used, each consisting of surface mass balance and 3D ocean thermal forcing. The climate
forcing follows the ISMIP6-Projections2300-Antarctica protocol (Seroussi et al., 2023) applied to our regional domain. The
two future climate scenarios considered come from UKESM (Sellar et al., 2020) – the one climate model used in ISMIP6-
Projections2300-Antarctica where both low- and high-emissions scenarios are available. Climate model structural uncertainty
is not considered in this study, and the two future scenarios used should be considered as broadly representative of high and
low greenhouse gas emissions scenarios.

For all the surface mass balance fields used, we add a large negative surface mass balance to land-based locations that are
ice-free in our initial condition to prevent ice advance into these areas, where basal friction is unconstrained. The associated
mass loss from this approximation is $< 2\%$ of the spatially integrated mass balance and is not a significant term in the overall
glacier mass balance. Surface air temperature, the upper boundary condition to ice thermodynamics, is kept at its historical
value throughout all simulations.

**Historical relaxation (RELX).** For each ensemble member, we conduct a 50-year relaxation run from the initial condition
using historical climate forcing to integrate out fast transient behavior. For surface mass balance, we apply a 1995–2017
climatological average from RACMO2.3p1 (Van Wessem et al., 2014; van den Broeke, 2019). The ocean thermal forcing
is the observation-based climatology compiled for ISMIP6-Antarctica, which uses data from 1995–2018 (Jourdain et al.,
2020; Nowicki et al., 2020). The 50-year relaxation duration is chosen as the most rapid adjustments occur in the first
few decades of integration, while the long-term adjustment to a fully steady state takes thousands of years. Relaxation to
full steady state requires substantially more computing resources than our entire set of ensembles, which would lead to
different runs having potentially very divergent initial states. Future improvements to model initialization that account
for surface elevation change (Perego et al., 2014) may reduce initial model drift and enhance this requirement. The
final model state in each run at the end of RELX is given the nominal date of January 1, 2015, and all three projection
ensembles are branched from these states.

**Control projection (CTRL).** The CTRL projection ensemble extends from the RELX configurations, continuing the same
surface mass balance and ocean thermal forcing from January 1, 2015 to January 1, 2300. This ensemble is used to
assess model drift relative to the forced response of the climate scenarios.

**SSP1-2.6 projection (SSP1).** Our SSP1 projection uses annual surface mass balance and ocean thermal forcing derived from
a UKESM SSP1-2.6 climate scenario (expAE10 from Seroussi and Nowicki, 2024). To avoid issues with climate model



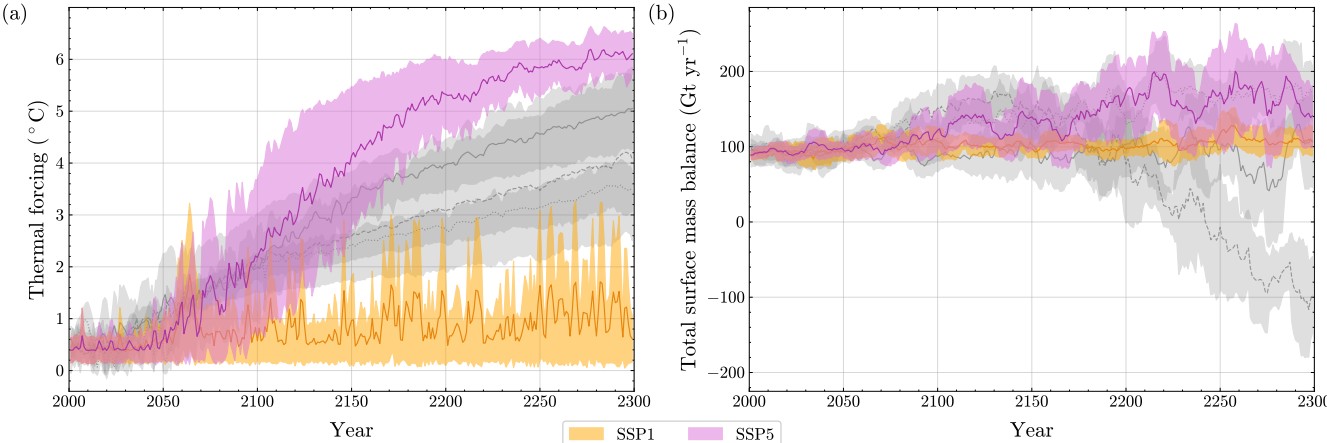

**Figure 2.** Time series of (a) thermal forcing averaged over the c. 2000 ice-shelf base for the UKESM SSP1 (orange) and SSP5 (purple) scenarios and three other RCP8.5 and SSP5 scenarios from other Earth system models provided by ISMIP6 (HadGEM2: gray-solid; CESM2: gray-dashed; CCSM4: gray-dotted) and (b) 10-year running mean total surface mass balance over the AmIS catchment. Shaded regions denote one standard deviation. (Seroussi and Nowicki, 2024).

bias and abrupt changes in forcing, surface mass balance and ocean thermal forcing are applied as anomalies relative to the climatological mean forcings in RELX/CTRL. The ensemble is run from January 1, 2015 to January 1, 2300.

**SSP5-8.5 projection (SSP5).** Our SSP5 projection uses the UKESM SSP5-8.5 projection forcings (expAE05 from Seroussi and Nowicki, 2024), again applied as anomalies and from 2015 to 2300.

## 3 Bayesian modeling framework

This section details the methodology used to facilitate the end-to-end, probabilistic modeling workflow featured in this work.

### 3.1 Perturbed parameter ensemble design

To quantify parametric uncertainty, MALI input parameters are varied to generate ensembles of MALI simulations. Here, we elaborate on the input parameters considered in this study, their prior distributions, and the sampling strategy used to generate
their values in the MALI simulations.

### 3.1.1 MALI parameters

The six MALI input parameters considered are summarized in Table 1. In Fig. 7, the green dashed lines represent prior probability distribution for each parameter.



**Table 1.** Summary of MALI parameters and their prior distributions.

| MALI Model Parameter | Sampled Range | Prior Distribution |
|---|---|---|
| Ice stiffness scaling factor, $C_\phi$ (Eq. 1) | (0.8, 1.2) | Truncated normal, center at 1.0, 2-sigma values at range limits |
| Basal friction scaling factor, $C_\mu$ (Eq. 2) | (0.8, 1.2) | Truncated normal, center at 1.0, 2-sigma values at range limits |
| Basal slip exponent, $q$ (Eq. 2) | (0.1, 0.333) | Trapezoidal, uniform likelihood between 0.15 and 0.28 |
| Calving yield stress, $\sigma_{max}$ (Eq. 3) | (80, 180) kPa | Truncated normal, center at 130, 3-sigma values at range limits |
| Ice-shelf melt coefficient, $\gamma_0$ (Eq. 4) | (9620, 471000) m yr$^{-1}$ | Truncated lognormal with $\log \gamma_0 \sim N(\mu, \sigma) = N(10, 1)$ |
| Ice-shelf basal melt rate, $\overline{m}$ (ice-shelf mean of $m$, Eq. 4) | (12, 58) Gt yr$^{-1}$ | Truncated normal, center at 35, 2-sigma values at range limits |

The ranges for ice stiffness scaling and basal friction scaling factors correspond to the values at which yield velocity solutions
exceed observational uncertainty and variability, based on sensitivity tests. For basal slip exponent, the high end of the sampled
range is the theoretically derived exponent for a hard bed (Weertman, 1957). The low end of $\frac{1}{10}$ is less than estimated exponents
for different Antarctic glaciers ($q < \frac{1}{5}$, Gillet-Chaulet et al., 2016) ($q = \frac{1}{9}$, Nias et al., 2018). The sampled range for calving
yield stress spans the range of values that produce approximately stable calving front positions in MALI for all major Antarctic
regions, based on sensitivity tests exploring model configurations for MALI's contribution to Seroussi et al. (2023). The low
end of the sampled range for ice shelf melt coefficient is the fifth percentile value for the nonlocal, Antarctic-wide (MeanAnt)
tuning from Jourdain et al. (2020), while the high end is the 95[th] percentile value for the high melt sensitivity tuning (PIGL)
derived from Pine Island Glacier. This spans the range of plausible values proposed by Jourdain et al. (2020) and used in
Seroussi et al. (2020). The last uncertain parameter identified in Section 2.1 is $\delta T$, the ocean temperature bias correction.
Because $\gamma_0$ and $\delta T$ are strongly dependent via Eq. 4, it is difficult to prescribe a range and prior distribution for $\delta T$, an ad-hoc
correction factor. Instead, we represent this degree of freedom using the uncertain historical ice-shelf averaged melt rate ($\overline{m}$)
itself. For a given sample of $\overline{m}$ and $\gamma_0$, we calculate the corresponding value of $\delta T$ to use for it. For $\overline{m}$, we use a normal
distribution with a mean and standard deviation corresponding to the AmIS as reported by Rignot et al. (2013), truncating at
their provided range and interpreting it as 2-$\sigma$ range around the mean which we also use to sample the $\overline{m}$ values.

### 3.1.2 Space filling sampling strategy

To achieve our objective of training a statistical emulator, we sample the parameter space uniformly within defined bounds
presented in Table 1. This approach helps to learn the model's response to varying inputs rather than relying on an expert-
defined distribution, which we later use as priors in Bayesian inference. We generate these uniform samples using a low-
discrepancy quasi-random Sobol' sequence (Sobol', 1967). Sobol' sequences possess uniform space-filling properties akin to
Latin hypercube sampling (McKay et al., 2000; Urban and Fricker, 2010), but they offer the advantage of recursively adding
new points while preserving their space-filling characteristics. This feature is particularly useful when expanding the ensemble
size later. In this study, we design a 200-member ensemble using this method (see Fig. A1).





## 3.2 Observations

We use three scalar observational constraints when performing Bayesian calibration of MALI parameters, specifically mass balance, grounding line movement, and calving front movement. While additional spatially resolved observations can be con-

215 sidered (e.g., ice surface velocity or elevation change rate), we choose to avoid the considerable complexity of weighting spatial misfit and combining spatial and scalar metrics. Instead, we restrict the observational criteria to large-scale scalar metrics. The modeled values of these observables are averaged over the 50-year RELX duration.

**Mass balance.** We use mass balance measurements of the grounded ice sheet from Rignot et al. (2019), which applied the input/output method from 1979–2017. We difference the 39-year average of surface mass balance (input) and discharge

(output) terms for the AmIS catchment, accounting for their stated uncertainties, to obtain a mean of $-1.656$ Gt yr$^{-1}$ and standard deviation of $5.720$ Gt yr$^{-1}$. Because these measurements are calculated from the input/output method, we compare the values to modeled grounded mass change rates using normal distribution likelihood $N(-1.656, 5.720)$ (instead of volume above flotation, which is an incomplete representation of mass balance).

**Grounding line movement.** Grounding line movement measurements from Konrad et al. (2018), estimated from satellite

altimetry collected between 2010 and 2016, are used in this work. We calculate the average grounding line velocity for the three glacier regions within the AmIS catchment (LAM, SCY, AME in Konrad et al., 2018), using the calculations for regions where ice flow speed exceeds 25 m yr$^{-1}$ and weighting by the length of grounding line captured in their surveys. While this is the most complete estimate available, the stated coverage for these three basins is 34-36%, and there are additional unsurveyed regions within the AmIS. By repeating the averaging for their 5[th] and 95[th] percentile estimates for

these regions and assuming errors are normally distributed, we obtain an estimate of grounding line movement with a mean of 0.467 m yr$^{-1}$ and standard deviation of 3.562 m yr$^{-1}$. For comparison with modeled glacier state, we multiply this grounding line velocity by the length of the grounding line in our domain (2117 km) to obtain a grounded area change rate of $0.988 \pm 7.540$ km$^2$ yr$^{-1}$, which we compare against modeled grounded area change rate using normal distribution likelihood $N(0.988, 7.540)$.

**Calving front movement.** Observations indicate that like other large Antarctic ice shelves, the AmIS has a multidecadal cycle of gradual advance followed by stepwise retreat through the detachment of large tabular icebergs (Fricker et al., 2002; Greene et al., 2022; Andreasen et al., 2023). At the same time, the advance and retreat cycle at AmIS has negligible impact on conditions in the central ice shelf (King et al., 2009) as these variations occur well past the portions of the ice shelf that provides significant buttressing (Fürst et al., 2016). Thus, the picture of the AmIS is one of stability with a

long-term front position remaining in the range of the advance and retreat cycle. The von Mises calving parameterization used cannot represent the advance and retreat cycles because it does not include the processes of damage and fracture formation that lead to tabular iceberg formation. Instead, it can only represent the long-term average calving behavior. Observations spanning 1963 to 2021 indicate calving front position has varied by 10,000 km$^2$ with the 2015 position about 1000 km$^2$ retreated from the most advanced position (Fricker et al., 2002; Greene et al., 2022; Andreasen et al.,





2023). To use these observations, we assume the range from the single observed AmIS calving cycle has a 50% likelihood of representing the true range of stable calving front variations and define a mean position of -4000 km$^2$ relative to the 2015 initial state with a standard deviation of 7353 km$^2$ (giving central 50% probability region of a normal distribution between -9000 and 1000 km$^2$). For our simulations, we then define long-term stability with the pragmatic criterion that the calving front position for the total simulation duration of 335 years (50 years RELX plus 285 years projection) should remain within the range of the observed calving cycle. Dividing the area change by the total simulation duration, we obtain a mean calving front position change rate of -11.94 km$^2$ yr$^{-1}$ with a standard deviation of 21.95 km$^2$ yr$^{-1}$. This observation-based estimate then is compared to modeled rate of change of the total ice area using normal distribution likelihood $N(-11.94, 21.95)$, which is a good proxy for calving front position change given the near absence of terrestrial and grounded marine margins in the catchment.

### 3.2.1 Ensemble filtering

Before training the RELX ensemble emulator and calibrating the parameters, we filter the runs in the RELX ensemble. The purpose is twofold. First, filtering eliminates outliers in potentially complex regions of parameter space that may have reduced the skill of the emulators but would be negligibly sampled based on their low likelihood of matching observations. Second, because in some cases our prior parameter distributions include regions of parameter space that will be negligibly sampled, eliminating runs from these regions reduces the computational cost of the three MALI projection ensembles. The applied filter removes runs that exceed four standard deviations of any of the three observational constraints (covering 99.9% of the central probability region around the mean in a normal distribution). This results in a filtered ensemble of 119 out of the original 200 runs being retained. A subset of these 119 runs are used for all subsequent statistical analysis.

## 3.3 Statistical emulation

The high-fidelity MALI ice sheet model simulations represent ice sheet dynamics accurately but are computationally expensive, prohibiting their use in generating a large enough ensemble of simulations for the purposes of uncertainty quantification. To address this challenge, we construct statistical emulators of MALI simulations that approximate MALI's behavior and outputs using statistical techniques (Sacks et al., 1989). Once constructed, the statistical emulators are used to explore the entire parameter space and capture the essential features of the MALI simulations at negligible computational cost. This also reduces the computational cost of quantifying uncertainty. While existing literature has demonstrated using various types of emulators, including regression analysis, Gaussian processes (GPs), neural networks, and machine learning algorithms (Berdahl et al., 2021; Bulthuis et al., 2019; Edwards et al., 2019), this study employs univariate and multivariate (multi-output) emulators, which are based on GP regression.

**Gaussian process emulators.** GP emulation (Gramacy, 2020) is a Bayesian nonparametric regression technique widely used to model complex systems. Unlike typical polynomial equations, GPs can capture nonlinear and nonparametric relationships without assuming a specific functional form. Moreover, by assuming the underlying process follows a Gaussian/ normal distri-





bution, GPs provide a flexible framework for capturing uncertainty and making predictions. GP emulation is a response-surface formulation (Box and Wilson, 1951) that treats the simulator as an unknown function of its input parameters (Gramacy, 2020). Inherently Bayesian, GPs express knowledge about unknown functions through probabilistic means, offering a robust method for modeling and prediction.

### 3.3.1 Gaussian process emulation of RELX ensemble

For this work, we construct separate scalar GP emulators to predict the three observables at the end of the 50-year relaxation using six MALI parameters in the RELX ensemble.

Each GP is constructed using a linear trend function and a separable Matérn covariance kernel with smoothness parameter ($\nu = 2.5$). Following standard practice, the hyperparameters controlling the mean function, as well as length scales and variance of the kernel, are optimized by minimizing the negative log-likelihood function. The pointwise-variance of the posterior prediction of the GP–referred to as *code uncertainty*–are used as a second independent noise term added to the scalar GP emulator's mean prediction during Bayesian calibration. The code uncertainty represents the error introduced by the emulator due to training on a small MALI ensemble (Kennedy and O'Hagan, 2001).

Each emulator is trained using input-output pairs obtained from the filtered 119-member ensemble and consisting of realizations of the six input parameters and corresponding scalar output. Both the input parameters and outputs are normalized to have unit $[0, 1]$ range and to improve the emulator training (we ensure the GP predictions are in the original output units).

**Emulator validation.** We train the emulators using 5-fold cross-validation. To investigate the accuracy of the resulting GPs, we plot ((Fig. 3(a), Fig. 4(a), Fig. 5(a))) the cross-validation residuals (the difference between the predicted and true outputs). The figures indicate that each GP emulator fits the simulation data quite well. Next, we plot the predicted versus actual values of scalar outputs, as well as the corresponding 50% predictive intervals, to assess the emulation skill of our GP models (Fig. 3(b), Fig. 4(b), Fig. 5(b)). In each figure, the emulator predictions are close to the 1-1 line, and predictive intervals are narrow, highlighting the good emulation skill with minimal uncertainty of the trained GP models. Subsequently, we plot the residuals against the pairwise input parameters to analyze trends associated with specific input parameter values leading to outliers in residuals (Fig. 3(c), Fig. 4(c), Fig. 5(c)). The figures do not show any significant patterns in the relationships between the outliers in the residuals and input parameters. Finally, we also plot predictive coverage plots, which helps assess how well the emulators are calibrated via predictive intervals (Fig. C7). As expected, perfect predictive coverage occurs when the model's predictive intervals align with the actual outcomes, and any deviation from this perfect coverage suggests the model's uncertainty estimates are either too conservative (*underconfident*: blue shaded region below dashed green line in Fig. C7) or too aggressive (*overconfident*: red shaded region above dashed green line in Fig. C7). Fig. C7 (a), (b), and (c), show that the GP emulators are slightly underconfident, overconfident, and slightly overconfident, respectively.





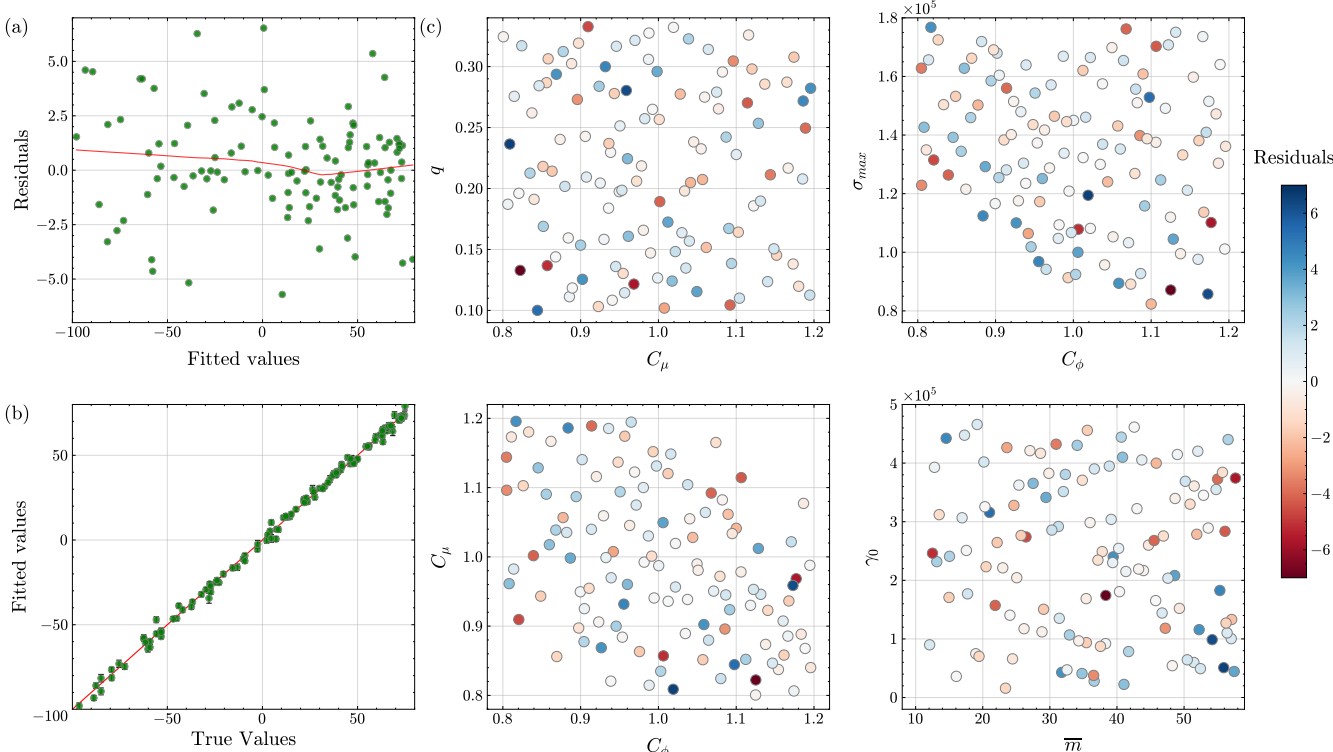

**Figure 3.** Analysis of Gaussian process emulator fit for calving front position: (a) residuals versus fitted values plot, (b) predicted versus actual values plot, and (c) residuals versus pairwise input parameters plots.

## 3.4 Bayesian calibration of MALI parameters

Once cross-validation of the GP emulators is complete, we train the emulators again on the full, filtered ensemble runs for the Bayesian calibration task. These GPs are then used to condition (or calibrate) the prior uncertainty of the MALI parameters on the three observations listed in Section 3.2. Specifically, following Kennedy and O'Hagan (2001), we update the prior probability distributions of the MALI parameters using Bayes' rule:

$$p(\boldsymbol{\theta}|\mathcal{D}) \propto p(\mathcal{D}|\boldsymbol{\theta})p(\boldsymbol{\theta}),$$

where $p(\boldsymbol{\theta})$ denotes the prior distribution of MALI parameters; $p(\mathcal{D}|\boldsymbol{\theta})$ represents the likelihood function, which is the joint probability of the observed data ($\mathcal{D}$) given the parameters $\boldsymbol{\theta}$; and $p(\boldsymbol{\theta}|\mathcal{D})$ is the posterior distribution of the MALI parameters. The prior distributions for the six MALI parameters, determined from past literature and previous model applications, are provided in Fig. 1. The likelihood function is assumed to be a multivariate Gaussian with a diagonal covariance $\boldsymbol{\Sigma} = \text{Diag}[\sigma_1^2, \sigma_2^2, \sigma_3^2]$ and satisfying

$$p(\mathcal{D}|\boldsymbol{\theta}) \propto \exp\left[-\frac{1}{2}(\boldsymbol{f}(\boldsymbol{\theta}) - \boldsymbol{d})^T \boldsymbol{\Sigma}^{-1} (\boldsymbol{f}(\boldsymbol{\theta}) - \boldsymbol{d})\right],$$





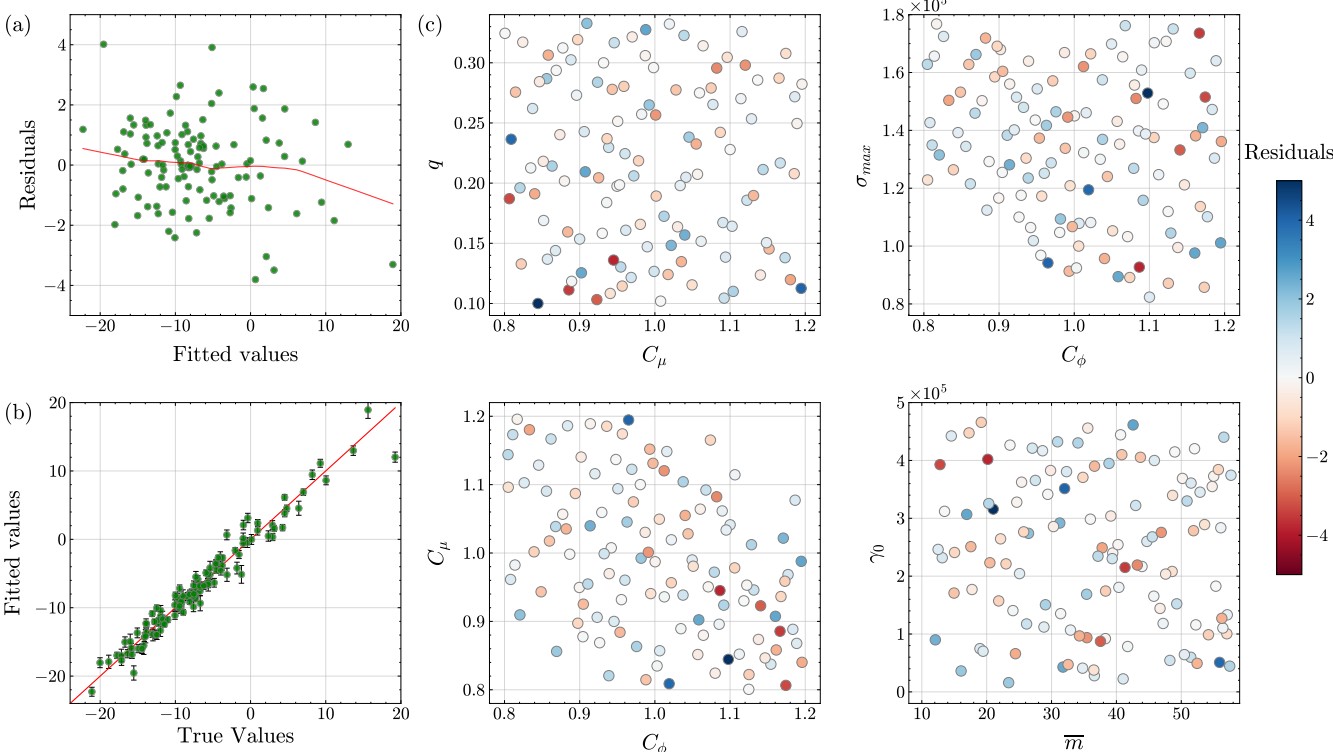

**Figure 4.** Analysis of Gaussian process emulator fit for grounding line position: (a) residuals versus fitted values plot, (b) predicted versus actual values plot, and (c) residuals versus pairwise input parameters plots.

which is equivalent to assuming that the noise in each of the three observations is independent. Here, $\boldsymbol{d}$ represents the vector of the three observations taken in 2015, and $\boldsymbol{f}$ is the vector of the three emulator outputs given $\boldsymbol{\theta}$. Additionally, we set the variance

term $\sigma_j^2$ for the $j^{\text{th}}$ scalar output as the sum of observational noise variance $\sigma_{o,j}^2$ and code uncertainty $\sigma_{c,j}^2$. The values of $\sigma_{o,j}$ are presented in Section 3.2. We then use the No U-Turn Sampler (NUTS) (Hoffman and Gelman, 2014), a Hamiltonian Monte Carlo method that requires minimal fine-tuning to draw samples from the analytically intractable posterior, $p(\boldsymbol{\theta}|\mathcal{D})$.

### 3.5 Principal component emulation of projection ensembles

This study's primary goal is to constrain and quantify the future contribution from the AmIS catchment to global mean sea

level using an ensemble of MALI simulations corresponding to draws from the posterior distribution of our input parameters. To achieve this, we calculate the sea-level contribution, from years 2015 to 2300, as the change in volume above flotation (VAF) converted to sea-level equivalent (SLE) following the definitions in Goelzer et al. (2020). We build and employ GP emulators to accelerate the probabilistic SLE projections for three different scenarios (CTRL, SSP1, and SSP5). The VAF in each of our projection ensembles (CTRL, SSP1, and SSP5) is time series data from 2015 to 2300. With high autocorrelation,

a multivariate (multi-output) emulator is more effective compared to individual scalar emulators for individual years, which is





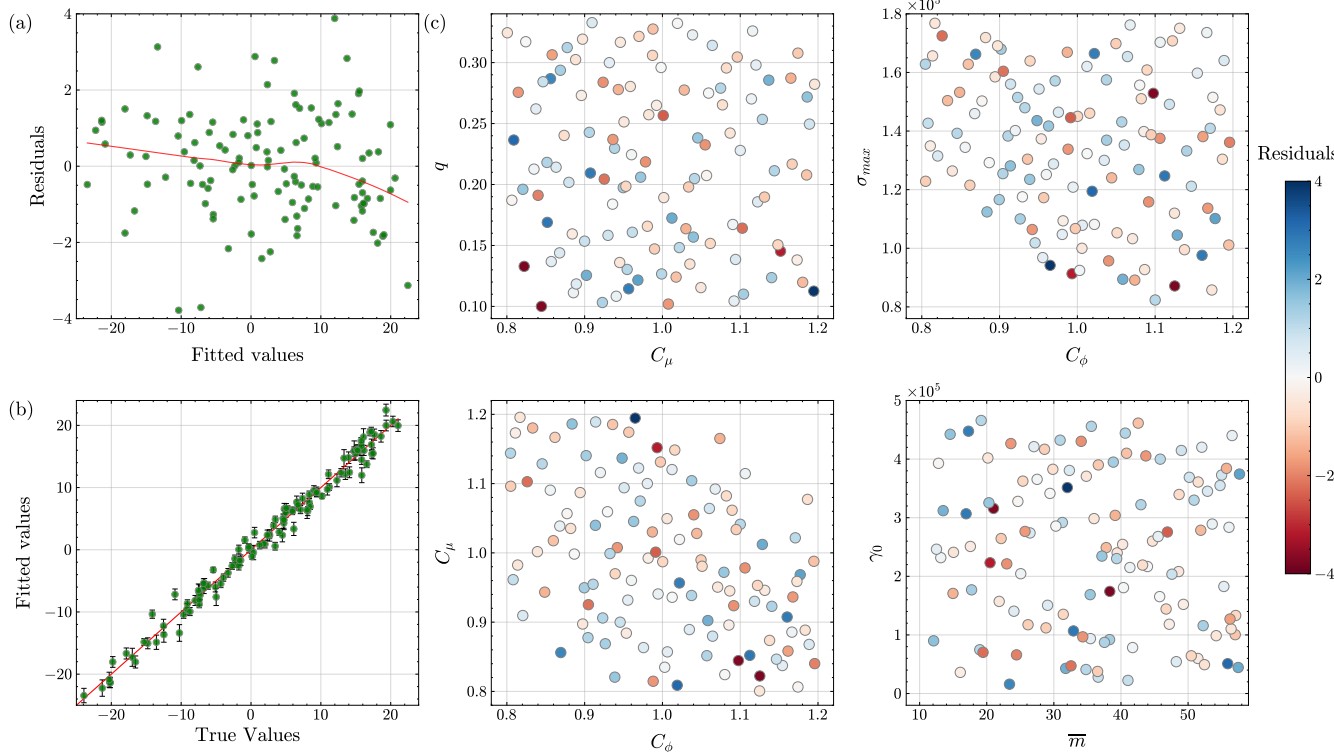

**Figure 5.** Analysis of Gaussian process emulator fit for mass change: (a) residuals versus fitted values plot, (b) predicted versus actual values plot, and (c) residuals versus pairwise input parameters plots.

cumbersome as additional treatment is required to account for autocorrelated data in consecutive years. To build a multivariate emulator, we employ principal component analysis (PCA) to extract linear reduced dimensional subspace of the time series similar to Higdon et al. (2008) and Wilkinson (2010). To this end, PCA provides projected data with maximized variance and uncorrelated but not necessarily independent dimensions.

We build the GP emulators using an ensemble of VAF change projections and the input parameter values in the filtered RELX ensemble. The resulting projection ensemble consists of 119 simulated time series data for VAF change over 285 years. Motivated by Higdon et al. (2008), PCA is used to reduce the dimensionality of this time series data from 285 to $K = 5$ principal components (a detailed description of these steps is available in Appendix B1). The value $K = 5$ is chosen because the retained principal components are able to explain $> 99\%$ of the variance in the original data. After this dimensionality reduction, we independently construct GP emulators to approximate the map between the input parameters and each of the retained principal components. When training the GPs, we transform the principal components and input parameters using unit [0,1] range transformations and again employ 5-fold cross-validation to validate the resulting PCA-based GP emulators. We reconstruct the mean and variance of the time series predictions from the $K$-emulated principal components using the procedure in Appendix B2.



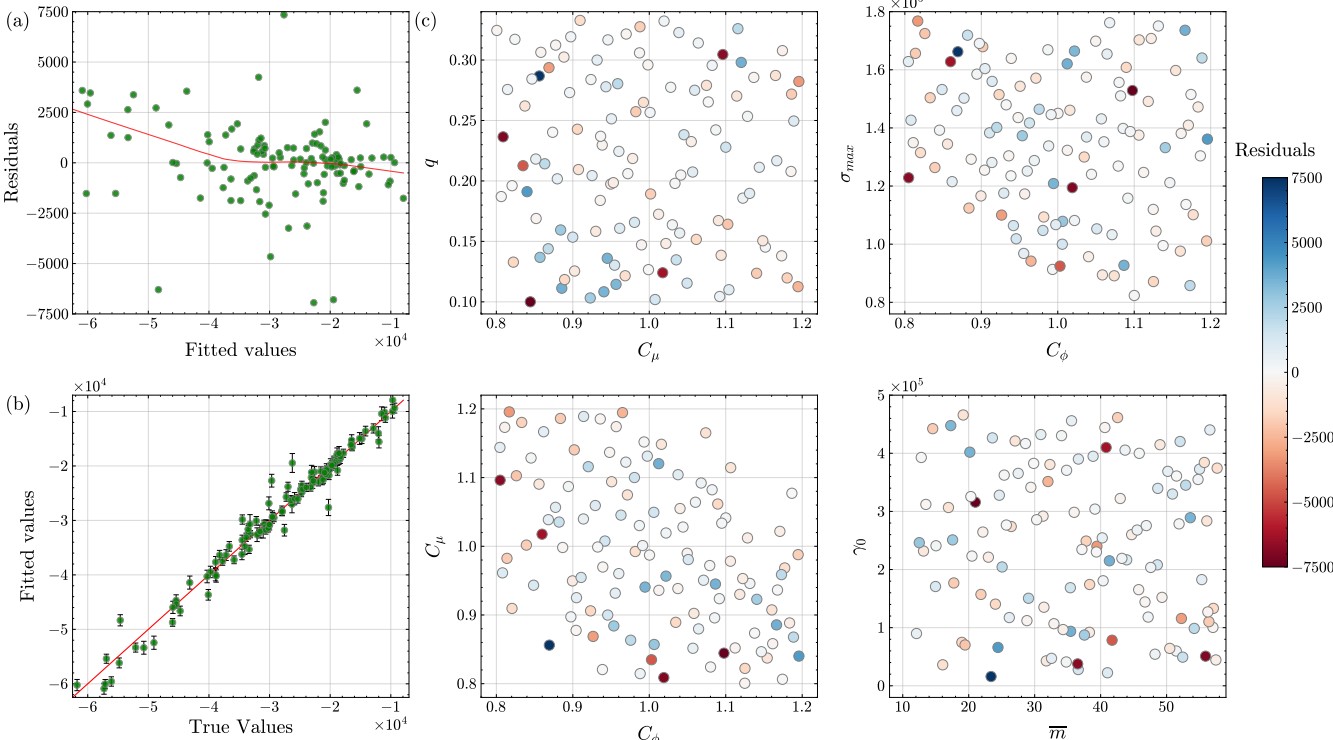

**Figure 6.** Analysis of multivariate PCA (five principal components) Gaussian process emulator fit for volume above flotation change in SSP5 projection ensemble at Year 2300: (a) residuals versus fitted values plot, (b) predicted versus actual values plot, and (c) residuals versus pairwise input parameters plots.

Fig. 6 summarizes the results of the PCA emulator 5-fold cross-validation for the SSP5 projection scenario at Year 2300. Similar results for CTRL and SSP1 projection ensembles are shown in Fig. C1 and C2, respectively. Fig. 6(a), shows no strong pattern in the residuals versus fitted values plot. Moreover, the PCA emulator predictions are close to the 1-1 line with narrow predictive intervals (Fig. 6(b)) and no specific patterns of residuals corresponding to inputs are observed (Fig. 6(c)). Lastly, the predictive coverage plot (Fig. C7(f)) shows good alignment of the model's predictive intervals with the expected outputs at Year 2300. We have analyzed years other than 2300 to assess the PCA GP emulator's performance but do not include those figures because, relative to the assessment at 2300, performance improves closer to the start of the time series. Finally, similar patterns to those described for SSP5 are evident in the CTRL and SSP1 projection ensembles, and their predictive coverage for these scenarios is summarized in Fig. C7(d) and (e).

Once the principal component emulators have been validated, we retrain them using the entire ensemble to provide probabilistic projections of future sea-level contributions from the AmIS catchment. We propagate the MALI parameter posterior samples through the GP emulators and generate samples from the emulator's predictive distribution for each of the $K = 5$





principal components. Finally, we reconstruct the sea-level contribution time series predictions from the principal components following steps detailed in Appendix B2.

# 4 Results

## 4.1 Bayesian calibration results

**Figure 7.** Posterior distributions of MALI parameters using Bayesian calibration on three observables: calving front position, grounding line position, and mass change, as well as their combined calibration effect. Prior distributions from Table 1 are shown using dashed green lines.

The posterior distributions of MALI parameters, calibrated to three observational constraints, are presented in Fig. 7. While most parameters are informed by the observations, certain observations inform some parameters more than others. Calibration using the calving front position (red lines) leads to posterior distributions that deviate from the prior distributions for calving





yield stress ($\sigma_{max}$) and ice stiffness scaling factor ($C_\phi$). The $\sigma_{max}$ posterior is symmetric but has shifted toward lower values

compared to the prior with a decreased variance indicating that values close to the median have a higher probability. The $C_\phi$

posterior is slightly skewed toward the lower values with the probability for values close to the median slightly increasing.

Calibration using grounding line position (blue lines) leads to posteriors that deviate from the priors for basal slip exponent

($q$), basal friction scaling factor ($C_\mu$), and ice-shelf basal melt rate ($\overline{m}$). Specifically, the posterior distributions for $C_\mu$ and $\overline{m}$

skew toward lower values, and the posterior for $q$ skew toward higher values. Additionally, $C_\mu$ and $\overline{m}$ posteriors have narrower

high-probability regions compared to their priors with values close to the median having a higher probability. Calibration using

mass change (yellow lines) leads to posterior distributions that deviate from the priors for the basal slip exponent, basal friction

scaling factor, ice stiffness scaling factor, and ice-shelf basal melt rate. The $q$, $C_\mu$, and $C_\phi$ posteriors are skewed towards lower

values, while the posterior for $\overline{m}$ is skewed towards higher values. $C_\phi$ has a narrower posterior distribution compared to its

prior, with values around the median having a high probability. In all three individual calibrations, the posterior for ice-shelf

melt coefficient ($\gamma_0$) does not change noticeably compared to its prior. In summary, each calibration leads to different posterior

distributions for the six MALI parameters, highlighting the impact and importance of each observational constraint.

Next, we detail the effect of simultaneously using all three observational constraints for calibration. As shown in Fig. 7

(purple lines), the posteriors for calving yield stress, basal friction scaling factor, ice stiffness scaling factor, and ice-shelf basal

melt rate deviate from their priors. For $\sigma_{max}$, the combination of three observables leads to a narrower posterior, centered at the

middle of the sampled range. For $C_\mu$, the posterior skews more toward lower values with a narrower high-probability region

than for the individual calibration corresponding to grounding line position and mass change. In addition, the $C_\mu$ posterior

peak shifts further left than for each individual calibration, possibly because combined calibration samples lower values more

frequently than higher ones from the range of possible values. For $C_\phi$, the combined calibration produces a posterior skewed

toward lower values with a higher peak and narrower high-probability region than the individual calibration based on calving

front position and mass change. In contrast, the $\overline{m}$ posterior is centered in the middle of the sampled range, suggesting that

skewness in opposite directions from calibrations corresponding to grounding line position and mass change cancel each other

out, amplifying the peak value and leading to a narrower distribution. Similarly, the $q$ posterior is similar to its uncalibrated

prior and has no skewness, suggesting that skewness in opposite directions from calibrations corresponding to grounding line

position and mass change cancel each other out. Lastly, as expected, the $\gamma_0$ posterior does not change noticeably from its prior

because all three individual calibrations leave the posterior for $\gamma_0$ close to its prior.

### 4.2   Probabilistic projection of future sea-level contribution from the Amery Ice Shelf catchment

Using the posterior distributions derived by calibrating all three observables, we generate 10,000 samples of the MALI pa-

rameters. Using the prior distributions given in Table 1, we also generate 10,000 samples of model inputs. These samples are

propagated through the PCA emulator for the SSP5 projection ensemble, generating the prior- and posterior-predictive sea-level

contribution time series, respectively. Fig. 8 summarizes the results. The posterior-predictive means are slightly larger than the

prior-predictive ones after incorporating all observational constraints. Moreover, the predicted uncertainties in the sea-level

contribution, shown by the 68% and 95% credible intervals, are narrower for the posterior than the prior with an increase in the





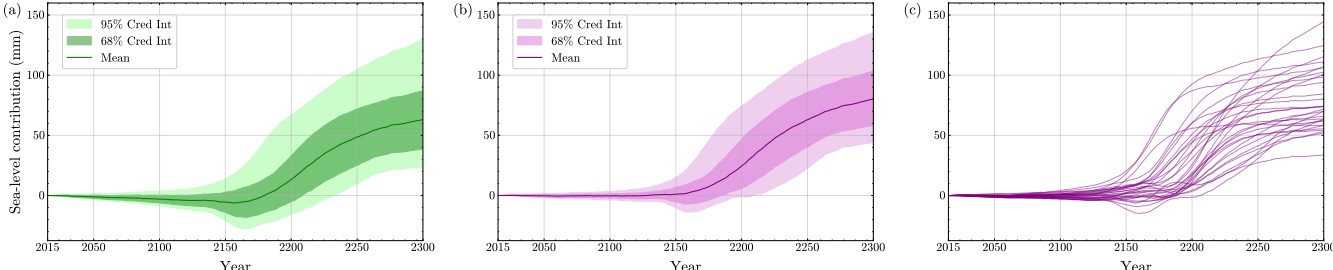

**Figure 8.** 10,000 samples, each from prior distributions and calibrated posterior distributions of MALI parameters, are propagated through the PCA emulator for SSP5 projection ensemble, leading to reconstructed time series for sea-level contribution (mm SLE) for years 2015–2300. (a) and (b): The mean and credible intervals at 68% and 95% around it using the prior (green) and posterior predictive (purple) sea-level contribution time series samples, respectively. (c): 30 randomly selected samples from the posterior predictive distribution.

lower bound of each interval. This underscores the importance of the Bayesian-calibrated ensemble over the RELX ensemble in terms of reducing MALI parameter uncertainties and their resulting impact on future sea-level rise uncertainty.

Next, we propagate the same set of posterior samples through the PCA emulators for the CTRL and SSP1 projection ensembles and generate posterior-predicted sea-level contribution time series. Fig. 9 shows a side-by-side comparison of probabilistic projections of future sea-level contribution in three scenarios: CTRL, SSP1, and SSP5. First, we observe there is a negative drift in the CTRL ensemble sea-level contribution projection because the initial conditions taken from the end of the RELX ensemble are not fully in equilibrium with historical climate. Second, we observe nearly identical trends and sea-level rise at

2300 in the SSP1 ensemble projection. Assuming that the drift toward a negative sea-level contribution is primarily due to an unequilibrated MALI transient in the control simulation, we can infer $\sim 0$ mm SLE future sea-level contribution in the low emissions scenario. Lastly, under the high emission (SSP5) scenario, the future sea-level contribution is estimated to be significantly higher than 0 mm SLE compared to both the control (CTRL) and low emission (SSP1) scenarios.

Finally, we generate future sea-level contribution predictions for specific years by emulating the individual year VAF change

MALI simulation data using a scalar GP emulator. Fig. C4, C5, and C6 respectively summarize the scalar GP emulator validation results using 5-fold cross-validation for the CTRL, SSP1, and SSP5 projection ensembles for Year 2300. In all three cases, there is no strong pattern in the residuals versus fitted values, the scalar GP emulator predictions are close to the 1-1 line with narrow predictive intervals, and no strong patterns in residuals corresponding to pairwise inputs is evident. Finally, the predictive coverage plots (Fig. C7 (g)-(i)) show good alignment of the model's predictive intervals with expected outcomes at

Year 2300 for all projection ensembles. Once our scalar GP emulators have been validated, we retrain them using the entire ensemble for a given year to provide probabilistic projections of future sea-level contributions from the AmIS catchment for the respective ensemble. We propagate the MALI parameter posterior samples through the GP emulator and sample from the emulator's predictive distribution. By combining these samples, we construct the posterior predictive distribution for the future sea-level contribution for a given year in each projection ensemble.



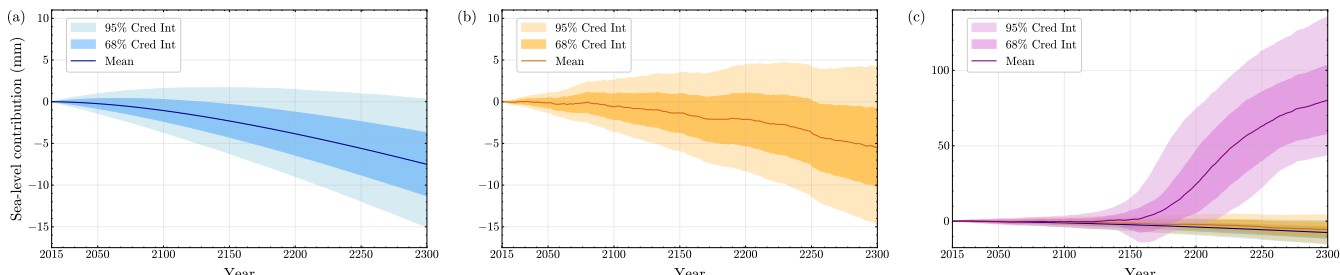

**Figure 9.** 10,000 samples from the calibrated posterior distributions of MALI parameters are propagated through the principal component emulators for (a) CTRL, (b) SSP1, and (c) SSP5 projection ensembles, leading to reconstructed time series for sea-level contribution (mm SLE) for years 2015–2300. We plot the mean and credible intervals at 68% and 95% around it using the posterior-predictive sea-level contribution time series samples. In (c), we include the plots for the CTRL (blue) and SSP1 (orange) ensembles to highlight the difference in sea-level contribution predictions from the SSP5 ensemble.

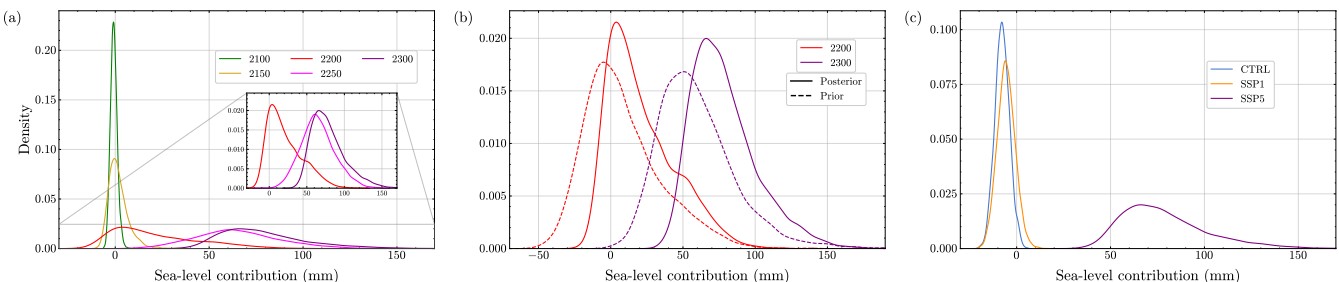

**Figure 10.** Posterior predictive distributions for the future sea-level contribution at representative years. (a) Posterior predictive distribution of future sea-level contribution for SSP5 scenario in the years 2100, 2150, 2200, 2250, and 2300. We zoom in on posterior predictive distributions of SLE in 2200, 2250, and 2300 to highlight their trends. (b) Prior predictive distribution (dashed lines) compared with corresponding posterior predictive distributions for future sea-level contribution for SSP5 scenario in 2200 and 2300. (c) Posterior predictive distribution of future sea-level contribution in Year 2300, corresponding to CTRL, SSP1, and SSP5 projection ensembles.

In Fig. 10(a), we plot the posterior predictive distributions of future sea-level contribution for the SSP5 emission scenario at years 2100, 2150, 2200, 2250, and 2300. We observe that sea-level contribution in 2100 and 2150 is negligible with -0.7 mm and 1.6 mm mean SLE. However, in 2200, the sea-level contribution is greater than 0 mm SLE with 21 mm mean SLE, and uncertainties in the distribution are wider. Moreover, in 2250 and 2300, the sea-level contribution is significantly higher than 0 mm SLE with 65 mm and 79 mm mean SLE, while Year 2300 has values that skew higher than in Year 2250,
suggesting greater future sea-level contribution. We also compare these posterior predictive distributions with the cross section of the posterior predictive time series distributions presented in Fig. 8 at years 2200, 2250, and 2300 in Fig. D1. As PCA is a lossy compression of data, we switch to individual year scalar GP emulators that are more accurate compared to the cross section of posterior predictive time series distribution at the chosen years. Fig. D1 shows slight differences between the posterior predictive distributions for the chosen years generated by the PCA and the scalar GP emulators. However, at Year





2200, there is a noticeable shift in the peak, which is expected since, as observed in Fig. C3, most of the sea-level contribution time-series exhibit sudden change between years 2150 and 2200. Next, Fig. 10(b) compares the prior and posterior predictive distributions for future sea-level contributions in Years 2200 and 2300. The posterior predictive means are 21 mm and 79 mm SLE, respectively, which are higher than prior predictive means of 8 mm and 59 mm SLE for those years. The 95% credible intervals (c.i.) of the posterior predictive distributions are (-9.6,75) mm and (46,133) mm SLE for Years 2200 and 2300,

respectively. In contrast, the prior predictive 95% credible intervals are (-31,70) mm and (19,123) mm SLE for the same years. This demonstrates that both the means and bounds of the 95% intervals increase while the intervals themselves shrink, indicating lower predictive uncertainties in future sea-level projections due to observational constraints for both years. Lastly, Fig. 10(c) presents the posterior predictive distributions of future sea-level contributions in the year 2300 for the CTRL, SSP1, and SSP5 projection ensembles. The posterior predictive mean and 95% c.i. for SSP1 ensemble are -5.5 mm and (-15,4) mm

SLE, which are quite similar to those for the CTRL ensemble at -7.7 mm and (-15,0.3) mm SLE. In contrast, the posterior predictive distribution for the SSP5 scenario indicates a significantly higher sea-level contribution in the year 2300, with a mean of 79 mm and a 95% c.i. of (46,133) mm SLE. This highlights a major sea-level contribution under the SSP5 scenario compared to the CTRL and SSP1 scenarios.

## 5 Discussion

### 5.1 Comparison to previous projections of the Amery Ice Shelf system

Previous modeling studies of Antarctica's AmIS sector suggest that the catchment is stable under the range of likely future climate with grounded ice mass loss occurring only with near complete loss of the ice shelf (Gong et al., 2014; Pittard et al., 2017). Barring ice-shelf removal, increases in surface mass balance (via increased accumulation in a warmer climate) may outweigh any acceleration in ice discharge caused by reduced ice-shelf buttressing. For scenarios where the AmIS remains

intact, Gong et al. (2014) project a mass change of 0 to -15 mm SLE at 2200, while Pittard et al. (2017) project -43 to 32 mm SLE (positive indicating sea-level rise). Given the differences in forcing scenarios and model configurations, our 95% c.i. for sea-level contribution in the SSP1 ensemble (where AmIS remains intact) of (-8.2,4.2) mm SLE at 2200 with -2.1 mm mean SLE is consistent with these earlier findings. We also point out that the 95% c.i. for SLE at 2200 in the CTRL ensemble is (-9,1.5) mm, where the mean SLE of -3.8 mm evolves from an expected mean of 0 mm, likely as a result of model drift.

Accounting for this drift in the CTRL ensemble, our SSP1 ensemble results suggest that, on average, we may not expect to see any significant changes in SLE as a result of future AmIS sector evolution. We attribute the narrower projection range in our study to considering a single, intact-AmIS scenario and, to a lesser extent, the observationally constrained calibration.

While these previous studies identify an increase in snowfall accumulation causing surface mass balance to rise, this effect is modest for the UKESM SSP1 atmospheric forcing used here, which yields a 14% cumulative increase in surface mass balance

at 2300 (relative to historical conditions (Fig. 2(b)). Furthermore, in our simulations, ice discharge increases in the SSP1 scenario, more than offsetting the small uptick in accumulation. Consequently, our SSP1 ensemble contributes slightly more





to sea-level rise than the CTRL ensemble (Fig. 9). Intermediate levels of climate warming are more likely to lead to increased accumulation, driving overall mass gain in the catchment (Pittard et al., 2017), but we have not evaluated such scenarios.

The SSP5 scenario applied here includes a sudden and sustained major increase in melting beneath the AmIS, leading to the

elimination of the ice shelf around 2130. This is caused by a sustained increase in mean thermal forcing not counterbalanced by the modest rise in surface mass balance (Fig. 2). Accordingly, the rapid mass loss in the second half of these simulations is depicted in Fig. 8. The 95% c.i. for our calibrated posterior distribution predicts (46,133) mm SLE during the final 170 years after the AmIS collapse. For scenarios where the AmIS is suddenly eliminated, Gong et al. (2014) predict up to 11 mm SLE in 200 years, whereas Pittard et al. (2017) forecast about 150 mm SLE after 170 years. It is important to note these previous

studies view the elimination of the AmIS as extreme and unlikely, whereas, in our simulations, it occurs as a consequence of state-of-the-art climate forcing and melt parameterizations (discussed further in Section 5.2).

Pittard et al. (2017) attribute the differences in sea-level contribution in their extreme scenario and that of Gong et al. (2014) to different bedrock topography datasets. The ALBMAP dataset (Le Brocq et al., 2010) used by Gong et al. (2014) includes a spurious ridge near the present-day grounding line, and the BEDMAP2 dataset (Fretwell et al., 2013) used by Pittard

et al. (2017) requires altering to deepen the bedrock near the grounding line by 500 m for consistency with oceanographic observations beneath the ice shelf. The BedMachine dataset, released after these studies, includes a trough along Mellor Glacier that is up to 2000 m deeper than that in BEDMAP2 (Morlighem et al., 2020; Morlighem, 2022) and is consistent with the modifications of Pittard et al. (2017). Thus, the consistency of our results with (Pittard et al., 2017) and the differences relative to Gong et al. (2014) appear to be due largely to the representation of bed topography. Similarly, Gong et al. (2014) report

grounding line stabilization within 200 years after ice-shelf removal, likely due to the relatively shallower bedrock topography. However, we see sustained grounding line retreat through the end of our simulations.

## 5.2 Ice-shelf collapse and response of grounded ice

Our results support previous studies that conclude substantial mass loss from the AmIS system is possible once the ice shelf is removed. However, while Pittard et al. (2017) predict the ice shelf will remain intact for the next 500 years, we show it

melting away within ∼130 years when forced by a state-of-the-art Earth system model under a high emissions scenario. While the UKESM SSP5 ocean forcing is the strongest of the forcings provided by ISMIP6, the other three Earth system models simulating the extended high emissions scenario agree on a strong influx of CDW within the next few centuries. Meanwhile, surface mass balance exhibits, at most, a modest increase (Figure 2), unlikely to offset increased ice discharge resulting from substantial ice-shelf thinning and retreat. The mean UKESM thermal forcing of ~3.5°C that causes ice-shelf loss in the 2130s

is reached under all three other SSP5 and RCP8.5 scenarios (Fig. 2). As significant ocean warming near the AmIS appears to be a robust feature of CMIP5 and CMIP6 models for high emissions scenarios, we expect that calibrated simulations of the AmIS forced by any of these models will predict a significant contribution to sea level by 2300. Furthermore, we include the potential impact of surface-melt-driven hydrofracture of the ice shelf (Scambos et al., 2009), which, if included, likely will accelerate the ice shelf's demise. Three of the four hydrofracture forcing maps provided by ISMIP6-AIS-2300 for SSP5 and





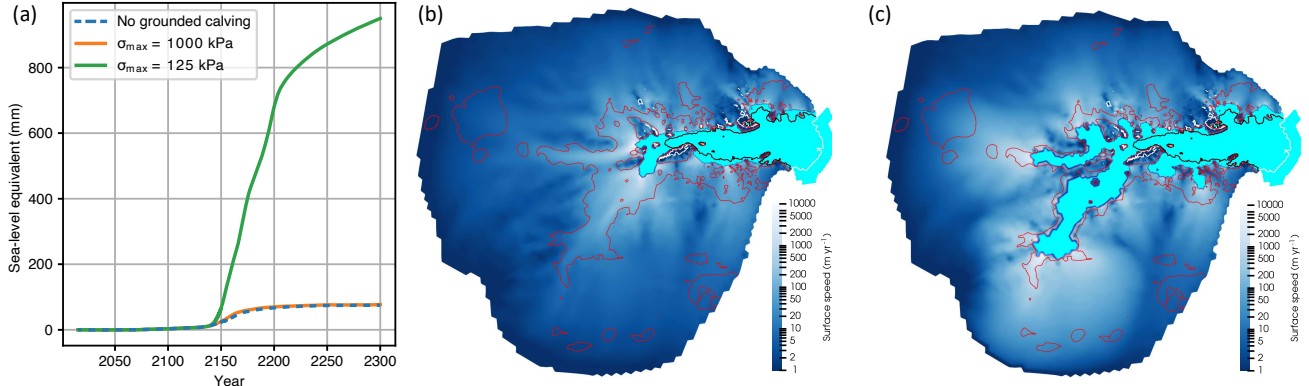

**Figure 11.** Sensitivity to treatment of grounded calving. a) Sea-level contribution when assuming no grounded calving (blue dotted line) as in primary results, using a typical value for tensile strength of grounded ice ($\sigma_{max} = 1000$ kPa, orange) and a value for tensile strength of ice close to the maximum a posteriori value for floating ice in our calibrated ensemble ($\sigma_{max} = 125$ kPa, green). b) Model state at Year 2300 when using $\sigma_{max} = 1000$ kPa. The white line indicates the initial calving front, and the black line is the initial grounding line. Red contours indicate where the bed topography is at sea level. Cyan shading indicates ice-free ocean portions of the domain. There is effectively no floating ice. c) Model state at 2300 when using $\sigma_{max} = 125$ kPa, as in (b).

RCP8.5 predict removal of the AmIS via hydrofracture between 2100 and 2200. While this process is highly uncertain and currently can only be crudely parameterized in ice-sheet models, it represents another potential threat to the AmIS' integrity.

     While our projections after ice-shelf removal in the SSP5 ensemble are consistent with the extreme scenario projections in Pittard et al. (2017), there is deep uncertainty about calving processes following the ice shelf loss. This stems from the lack of observations against which to calibrate models and the poorly understood physics (Pollard et al., 2015). As described in

Sec. 2.1, we have disabled calving at grounded marine termini because no such termini exist for this glacier system under historical conditions, preventing the possibility of calibrating associated parameters. To explore the impact of this modeling choice for a typical ensemble member, we perform sensitivity experiments, enabling calving at grounded marine termini.

     To select a typical run, we first identify the ensemble members from the filtered SSP5 projection ensemble with mass change at Year 2300 between the 40th and 60th percentiles. Of this subset, we identify one run with parameter values for the calving

yield stress (125,158 Pa), basal slip exponent (0.2245), basal friction scaling factor (0.9602), and ice stiffness scaling factor (0.9601), which are all close to the means of the respective posterior distributions. This run has a mass change at 2300 of 77 mm SLE, which is close to the mean of the sea-level contribution's posterior predictive distribution at 2300 of 79 mm SLE (Fig. 10(a)). Using this set of parameter values, we conduct additional SSP5 simulations with the calving yield stress for grounded marine termini set to 1000 kPa, a typical value used for the tensile strength of ice (Petrovic, 2003; Morlighem et al.,

2016; Ultee et al., 2020), and a more aggressive threshold set to the value used for floating ice (125 kPa).

     Including grounded calving with a threshold stress of 1000 kPa leads to a slightly faster mass loss after the removal of AmIS, but the glacier mass stabilizes at about the same value as when ignoring grounded calving (Fig. 11(a)). However, using the





same 125 kPa threshold stress as for floating ice leads to a tenfold increase in mass loss, yielding nearly 1 m SLE at Year 2300. This behavior is analogous to the Marine Ice Cliff Instability (Bassis and Walker, 2012; Pollard et al., 2015; DeConto and Pollard, 2016). As implemented by Pollard et al. (2015); DeConto and Pollard (2016), the Marine Ice Cliff Instability is a calving rate that is a function of ice cliff height above the water line. Here, the analogous behavior comes from the calving rate being a direct function of effective tensile stress. The effect is similar in that large, effective tensile stress yields large calving rates and grounded margin retreat, which, in turn, leads to larger effective tensile stresses and faster retreat when the margin withdraws into deeper bedrock (Fig. 11(b) and (c)), because the effective tensile stress is strongly affected by subaerial cliff height. The margin retreat and mass loss seen here with a grounded threshold stress of 125 kPa is roughly similar to that shown by DeConto and Pollard (2016) for this same region.

The inability to calibrate the calving yield stress for grounded marine margins complicates future projections for the glacier system under this configuration. The tensile strength of ice has been estimated at an order of 1 MPa (Bassis and Walker, 2012; Ultee et al., 2020). The fact that our calibrated value for floating ice (125 kPa) is so much lower may represent unresolved damage and crevassing (Bassis and Ma, 2015; Lhermitte et al., 2020; Kachuck et al., 2022) in our model, which would lead to a yield strength lower than that for intact ice. In this case, a higher threshold stress would be appropriate for grounded ice than for floating ice because the former is typically less damaged than the latter (Bassis and Ma, 2015; Kachuck et al., 2022). The inability to calibrate this parameter implies its especially large uncertainty with respect to the representation of grounded ice calving in models. Notably, until relevant direct observations or improved prior understanding are available, this uncertainty will continue to limit the utility of Bayesian inference for quantifying the uncertainty in future sea-level rise from Antarctica.

### 5.3 Quantification of parametric uncertainty

A notable result of our Bayesian calibration is that the observables constrain both the MALI parameters (Fig. 7) and the projections of the MALI model, as observed by comparing the posterior and prior predictive distributions of future sea-level contribution (Fig. 8, 10(b)). Combining three observational constraints leads to calibrated posteriors for the calving yield stress, basal friction scaling factor, ice stiffness scaling factor, and ice-shelf basal melt rate, which differ from their uncalibrated priors, whereas the posteriors for the basal slip exponent and ice-shelf basal melt rate remain similar to their priors (Fig.7). Consequently, when the posterior samples of MALI parameters are propagated through the scalar GP emulator of SSP5 projection ensemble for the Year 2300, they yield a posterior predictive distribution of future sea-level contribution with a higher mean of 79 mm SLE (Fig. 10(b)). In contrast, prior samples of MALI parameters result in a prior predictive distribution with a mean of 59 mm SLE for the same year. Moreover, the posterior and prior predictive 95% credible intervals are (46,133) mm and (19,123) mm SLE, respectively, at Year 2300. This highlights that both mean and the bounds of the credible interval have increased, while the interval itself has reduced in size for the posterior predictive distribution. This indicates lower predictive uncertainties in future sea-level projections due to the calibrated MALI parameter samples. These observations underscore the importance of Bayesian calibration of MALI parameters in constraining and quantifying the future-sea level contribution uncertainties. Potential avenues for reducing uncertainty include: (1) collecting new observations with reduced uncertainty, (2) evaluating additional observations (either our existing scalar metrics or additional metrics) at multiple time epochs, or (3)





adding new scalar or spatially resolved quantities of interest. Applying Bayesian calibration to the relatively stable AmIS system with effectively a single time point of observations limits the amount of information that it can provide, as scant information about the dynamical response to external perturbations is included in the calibration. The deep uncertainty associated with calving at grounded marine margins is an extreme case of lack of observations — observations relevant to the key processes must exist to derive full benefit from the calibration. Incorporating paleo data (Gilford et al., 2020) or additional locations where these processes operate under historical conditions afford additional paths forward.

## 5.4 Model limitations and future work

There are a number of potential improvements to the ice-sheet model configuration used here. The 4-km resolution used near the grounding line is likely too coarse to fully resolve grounding line dynamics (Gong et al., 2014; Hoffman et al., 2018). This is a computational requirement for running a large enough ensemble to support the generation of emulators. Similarly, the model spin-up procedure results in model drift of up to about -10 mm SLE (Fig. 9(a)). Conducting multi-millennia spin-ups for each ensemble member (e.g., Berdahl et al., 2023) or improving model initialization techniques to minimize drift (e.g., Perego et al., 2014) would reduce this issue. We note that both approaches introduce other complications through increased simulation cost and by additional differences in the initial state between ensemble members.

There are also improvements to be made to the physical process representations in the model. Although the von Mises stress calving law employed performs well at simulating a stable AmIS calving front (Wilner et al., 2023), it cannot reproduce the historical cycles of calving front retreat and advance (Fricker et al., 2002; Greene et al., 2022; Andreasen et al., 2023). More sophisticated methods that can reproduce these cycles have yet to be implemented in large-scale ice-sheet models, but this is an area of active research (Bassis et al., 2024). The assumption of a time invariant friction coefficient is also a limitation that requires implementation of new process models to resolve. We also neglect glacial isostatic adjustment, which can have strong feedbacks with marine ice-sheet evolution (Gomez et al., 2010) but likely relatively minor impacts on the timescales considered here and in this region, where mantle viscosity is high (Whitehouse et al., 2019; Coulon et al., 2021). While we present probabilistic projections, we only consider two climate scenarios, and the climate scenario uncertainty is far larger than the parametric uncertainty being considered. Our use of a single Earth system model for climate forcing also presents a limitation necessitated by computational expense. For a given SSP, there is large uncertainty in ice-sheet model projections when applying climate forcing from different climate models (Seroussi et al., 2020, 2023; Seroussi and Nowicki, 2024).

## 5.5 Limitations and potential improvements to uncertainty quantification methods

Any data-model comparison is limited by the number and dimension of both input uncertainties and output data constraints. This study considers only six scalar inputs as uncertain variables to be estimated. We judge these to be influential based on domain expertise, but the model contains other parameters whose values are not known. In particular, the ice stiffness and basal friction scaling factors are a highly simplified means to represent uncertainties in what are actually two-dimensional spatial fields. Uncertainty quantification of high-dimensional field data is an open mathematical research challenge (Isaac et al., 2015; Petra et al., 2014; Brinkerhoff, 2022; Hartland et al., 2023; Riel and Minchew, 2023; Reese et al., 2024) often resorting to



Gaussian posterior or other approximations to accomodate the computational expense. Another open technical question is how to combine such approximate methods with the fully non-Gaussian, asymptotically convergent Markov chain Monte Carlo methods used here for parameter estimation.

An additional limitation of this work is its reliance on scalar, temporally and spatially averaged observational constraints. We observe that the basal slip exponent ($q$) and the ice-shelf melt coefficient ($\gamma_0$) are largely unconstrained by Bayesian
calibration, which suggests that our observational metrics are not informative about these parameters. This may be a result of over-averaging data in space and time, which can remove information. We could consider space- and time-resolved data (where available), although this will introduce additional complexities in modeling the data-model residual spacetime covariance structure. This structure is likely intricate, and statistical misspecification of the model discrepancy (error) can bias inference. Calibration against multivariate data using a dimensionally reduced principal component emulator also can introduce bias
(Salter et al., 2019; Salter and Williamson, 2022). We can consider including additional observational constraints. Felikson et al. (2023) demonstrates the choice of observable can significantly impact the projections of ice-sheet mass change. Finally, a more sophisticated treatment of multiple observational constraints will require careful treatment of the statistical covariance between observables (or, rather, between the model errors for different observables).

## 6    Conclusion

This study provides a comprehensive analysis of the future sea-level contribution from the AmIS catchment with quantified uncertainties. Three different observables – mass balance, grounding line movement, and calving front movement – are chosen to effectively calibrate the input parameters to the MALI ice-sheet model. Statistical emulation using GPs is used to significantly reduce the computational burden of performing Bayesian calibration and uncertainty propagation. Each observable leads to individual Bayesian calibration of MALI parameters using their expert-knowledge-based priors, resulting in posterior
distributions. Next, we combine the effects of these three observables to effectively capture final calibrated posterior distributions of MALI parameters. The combined calibration helps us constrain the calving yield stress, basal friction scaling factor, ice stiffness scaling factor, and ice-shelf basal melt rate parameters, while basal slip exponent and ice-shelf basal melt rate remain unconstrained. The calibrated posteriors and respective priors are used to sample MALI parameters to propagate them through the multivariate principal component emulator to obtain posterior and prior predictive distributions of future sea-level
contribution from AmIS under two different emission scenarios.

Using the SSP5 high emission scenario to highlight changes in this relatively stable glacier, our expert prior over the MALI parameters, when propagated through our scalar GP emulator, projects a mean SLE of 59 mm with a 95% c.i. of (19,123) mm in the year 2300. After constraining the MALI model parameters with observations, this projection changes to a mean SLE of 79 mm with a 95% c.i. of (46,133) mm. Both the mean and the bounds of the 95% interval increase, while the interval
itself shrinks, indicating lower predictive uncertainties in future sea-level projections due to observational constraints. The reduction in future sea-level contribution uncertainties is modest, due to the limited quantity of observations and their inherent uncertainties for the glacial system. In addition, because this system is close to equilibrium, the dynamical constraints on retreat





are weak. While the reduced complexity resulting from the AmIS catchment being close to equilibrium makes this a relatively straightforward application of the methodology, application to other more dynamic glacier systems, such as Thwaites or Totten

Glaciers, potentially will lead to better informed projections. We develop a framework providing an end-to-end probabilistic modeling workflow consisting of Bayesian calibration and uncertainty propagation together with Gaussian process emulation for similar studies that can be undertaken in the glaciological community and beyond.

While Bayesian calibration is expected to be more informative for situations with a larger range of observed behavior, our calibrated projections allow for assessing how parametric uncertainty affects the projected ice-sheet response to abrupt

changes in oceanic forcing. The high and low greenhouse gas emission scenarios yield similar projections of glacier mass change for the twenty-first century. After abrupt ocean warming and elimination of AmIS through basal melting, the difference between the scenarios becomes statistically significant within decades, far exceeding the parametric uncertainty at Year 2300. The results align with previous projections of the AmIS catchment, indicating stability with minimal contribution to sea-level change *unless* large changes in ocean temperature occur, leading to substantial sea-level rise. What has changed since previous

studies of the AmIS system is the subsequent generation of CMIP model projections to 2300, predicting large ocean warming under high greenhouse gas emission scenarios, previously considered unlikely. Our study uses these newer climate projections while quantifying parametric uncertainty, allowing for evaluating the significance of differences between projections of the AmIS catchment under these climate scenarios. Major remaining uncertainties include the processes of calving and marine melting once the ice shelf is removed. Feedbacks between stress state and calving at marine cliffs potentially can lead to

mass loss 10 times larger than our primary ensemble, leaving a deeply uncertain long tail to future projections. Reducing this deep uncertainty will require applying uncertainty quantification methods to locations where these processes have been observed over a diverse range of behavior and gaining fundamental understanding of calving processes to confidently transfer this knowledge to places like AmIS, where such events have not yet occurred.



## Appendix A: Input parameters



**Figure A1.** Sobol' sequence generated 200 MALI simulation runs input parameter initializations.





## Appendix B: PCA emulator details

### B1   Principal component analysis dimensionality reduction steps

This appendix presents the detailed steps for the dimensionality reduction of time series data using principal component analysis (PCA). We begin with $T \times N$ time series data matrix denoted by $S$ and extract reduced K-dimensional principal subspace, $Z_1$, which has uncorrelated rows using the linear transformation matrix, $P_1$, consisting of K eigenvectors corresponding to top K eigenvalues using following steps:

1. Compute the vector $\mu$ of row means of the data matrix $S$. Center $S$ by subtracting $\mu$ from each column and get $S_c$.

2. Find the singular value decomposition of $S_c$:

$$S_c = U\Lambda V^T,$$

   where $U$ is $T \times T$ orthogonal matrix of left singular vectors of $S_c$. $V$ is $N \times N$ orthogonal matrix containing the right singular vectors of $S_c$. $\Lambda$ is a $T \times N$ diagonal matrix containing singular values arranged in a descending order and large singular values correspond to the important feature directions in the $S_c$ (Shlens, 2014).

3. Let $P = U^T$ then we see that

$$PS_c = U^T S_c = \Lambda V^T = Z.$$

   The rows of $P$ represent principal components (PCs) of $S_c$. $Z$ represents the fully transformed data matrix (or full principal subspace) using all of the PCs.

4. Choose dimensionality of the reduced principal subspace and denote it by $K, (K < T)$. Let $P_1$ consist of the first $K$ principal components then $P_1$ forms the orthonormal basis for the reduced principal subspace. Next, project $S_c$ onto the reduced principal subspace using the $K \times T$ transformation matrix $P_1$ as follows:

$$P_1 S_c = Z_1,$$

   where $Z_1$ represents the reduced-dimensional, $K \times N$, transformed data matrix.

### B2   PCA emulator predictive mean and variance construction

We present the steps for reconstruction of mean and variance of the time series predictions from $K$ emulated PCs. First, we back-transform $\hat{Z}_1$, the predictive means of Gaussian process (GP) emulators of K PCs row-wise stacked together, as follows:

$$\hat{S}_c = P_1{}^T \hat{Z}_1.$$

Then, we decentralise $\hat{S}_c$ by adding row means, $\mu$, of the data matrix $S$ to obtain $T \times N$ matrix $-\hat{S}$ representing mean of time series predictions. Next, we reconstruct variance by first reconstructing the entire covariance matrix of each time series prediction as follows:

$$\mathcal{C}_t = P_1{}^T \mathrm{diag}(\sigma_{1,t}^2, \cdots, \sigma_{K,t}^2)P_1,$$





where $\mathcal{C}_t$ represents covariance matrix for the $t^{\text{th}}$ time series prediction and diagonal entries are variances. Next, we row-wise concatenate individual time series variance predictions to get $T \times N$ matrix $\Sigma$. We use $\Sigma$ to evaluate credible intervals around the mean predictions, $\hat{S}$.



## Appendix C: Emulator validation continued

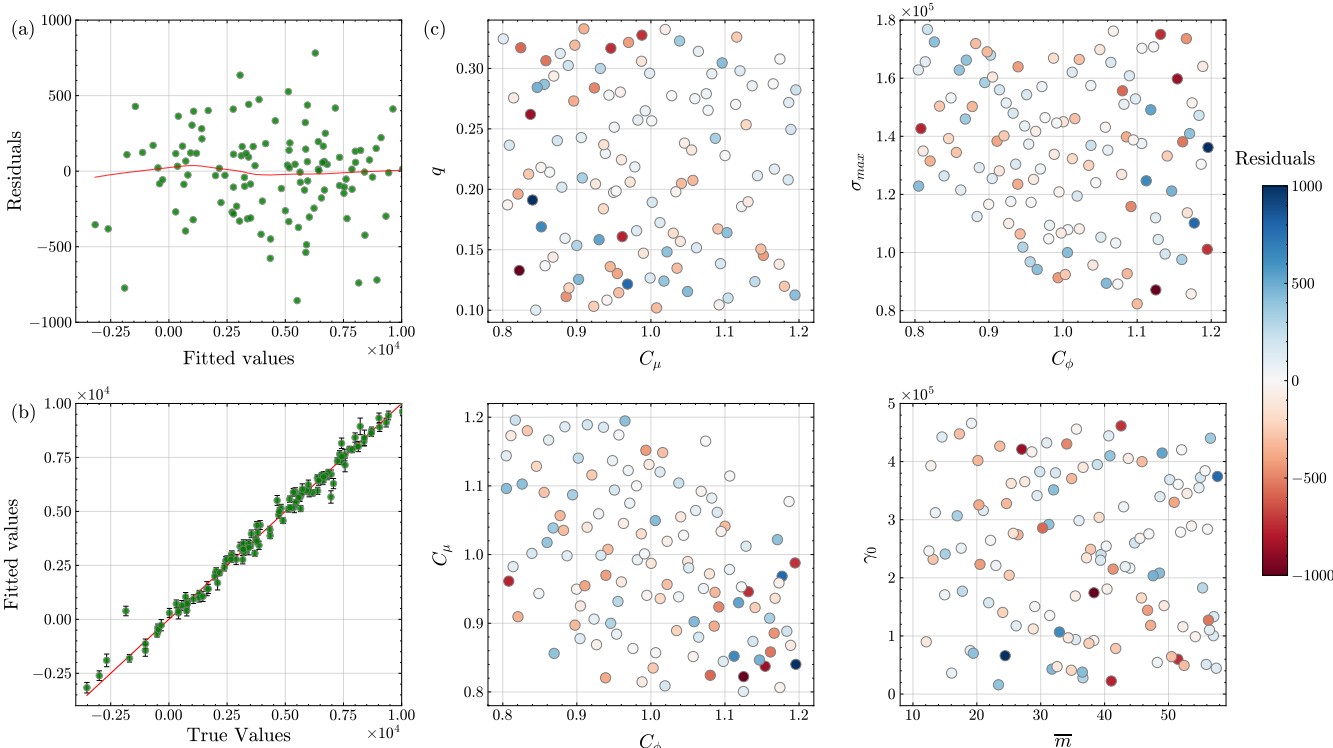

**Figure C1.** Analysis of multivariate principal component analysis (PCA) (five principal components) GP emulator fit for volume above flotation change in CTRL projection ensemble at Year 2300: (a) residuals versus fitted values plot, (b) predicted versus actual values plot, and (c) residuals versus pairwise input parameters plots.





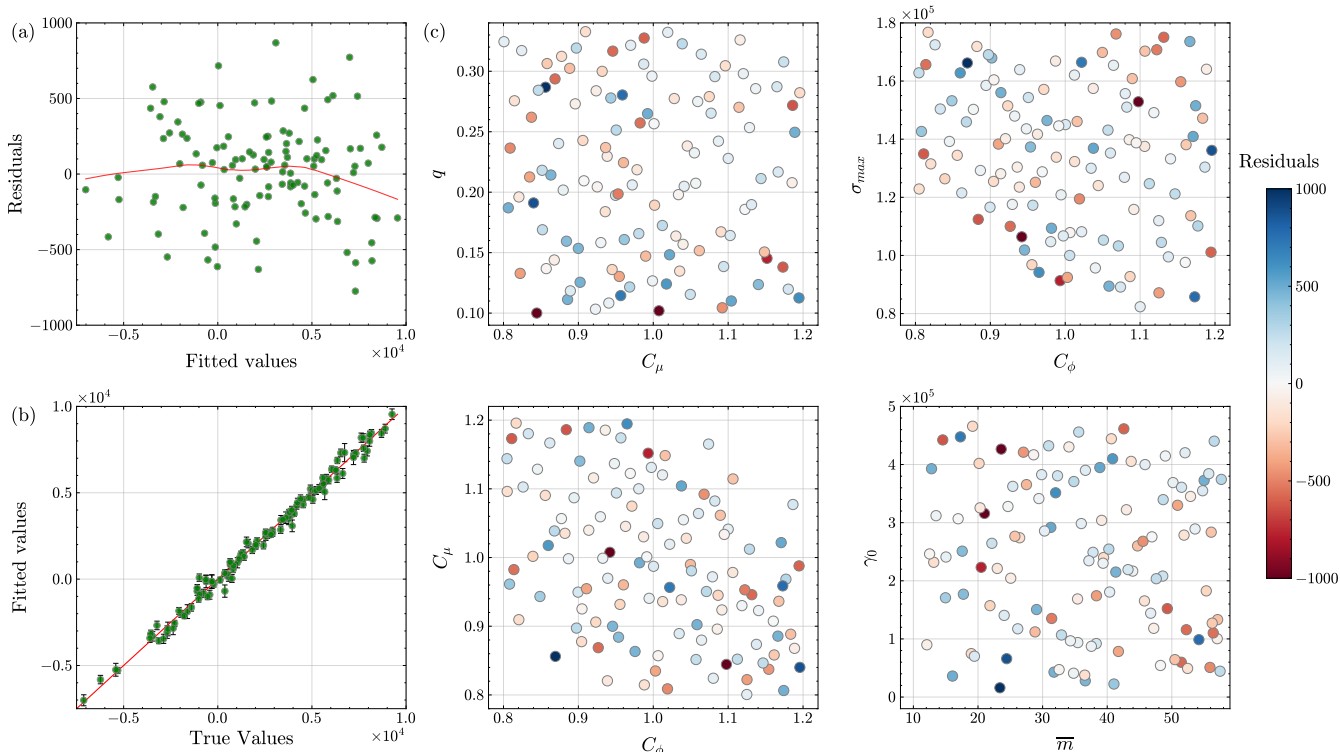

**Figure C2.** Analysis of multivariate PCA (five principal components) GP emulator fit for volume above flotation change in SSP1 projection ensemble at Year 2300: (a) residuals versus fitted values plot, (b) predicted versus actual values plot, and (c) residuals versus pairwise input parameters plots.



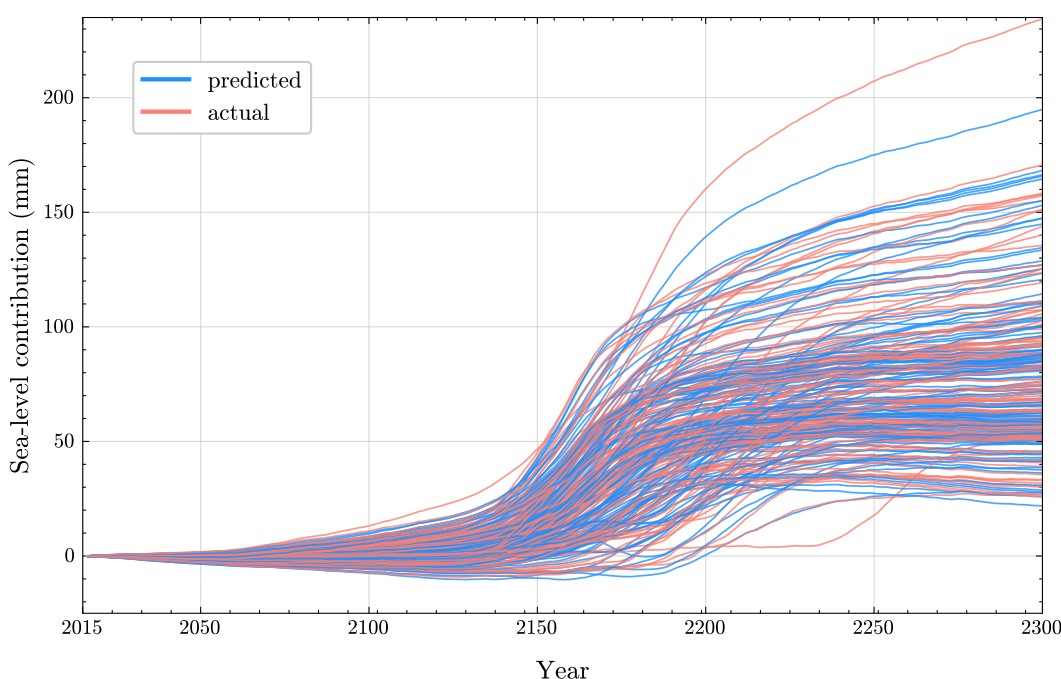

**Figure C3.** Multivariate PCA (five principal components) GP emulator predictions during 5-fold cross-validation compared with the actual data for sea-level contributions (in mm) until the Year 2300.





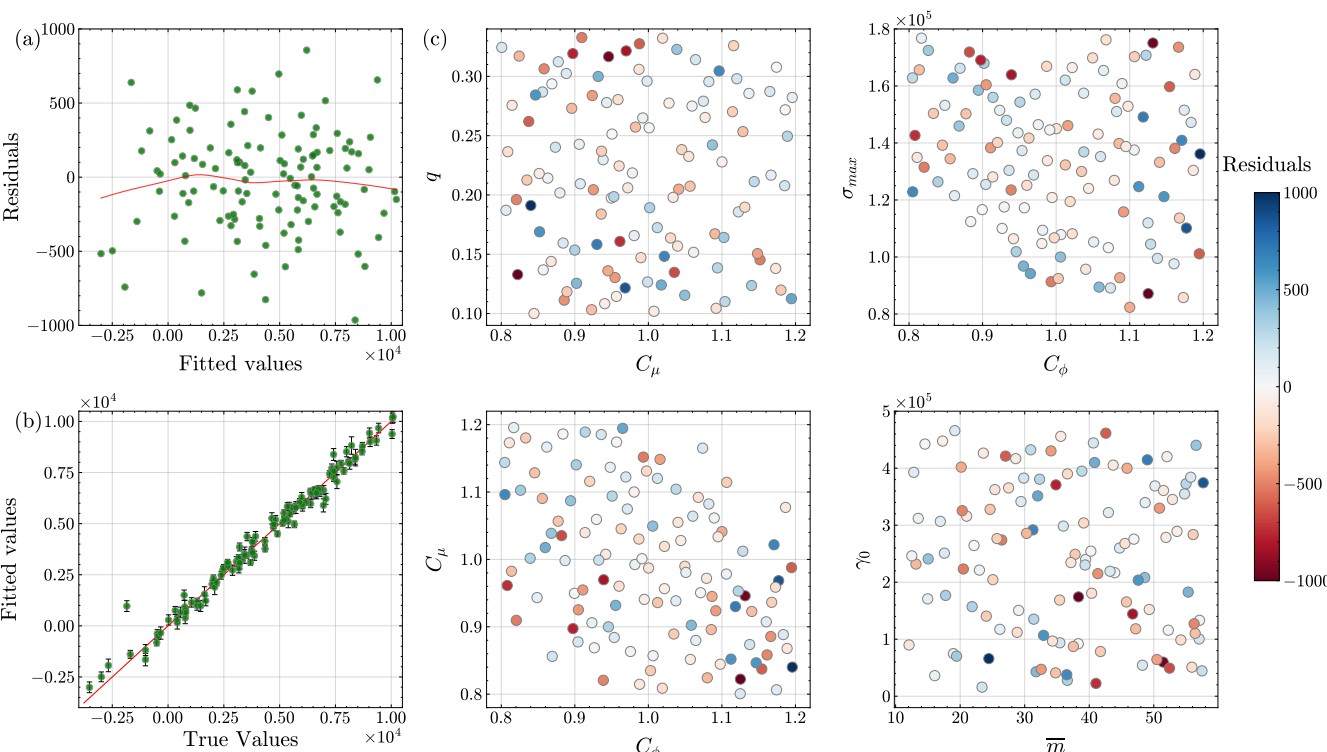

**Figure C4.** Scalar GP emulator fit analysis for volume above flotation change in CTRL projection ensemble at Year 2300: (a) residuals versus fitted values plot, (b) predicted versus actual values plot, and (c) residuals versus pairwise input parameters plots.

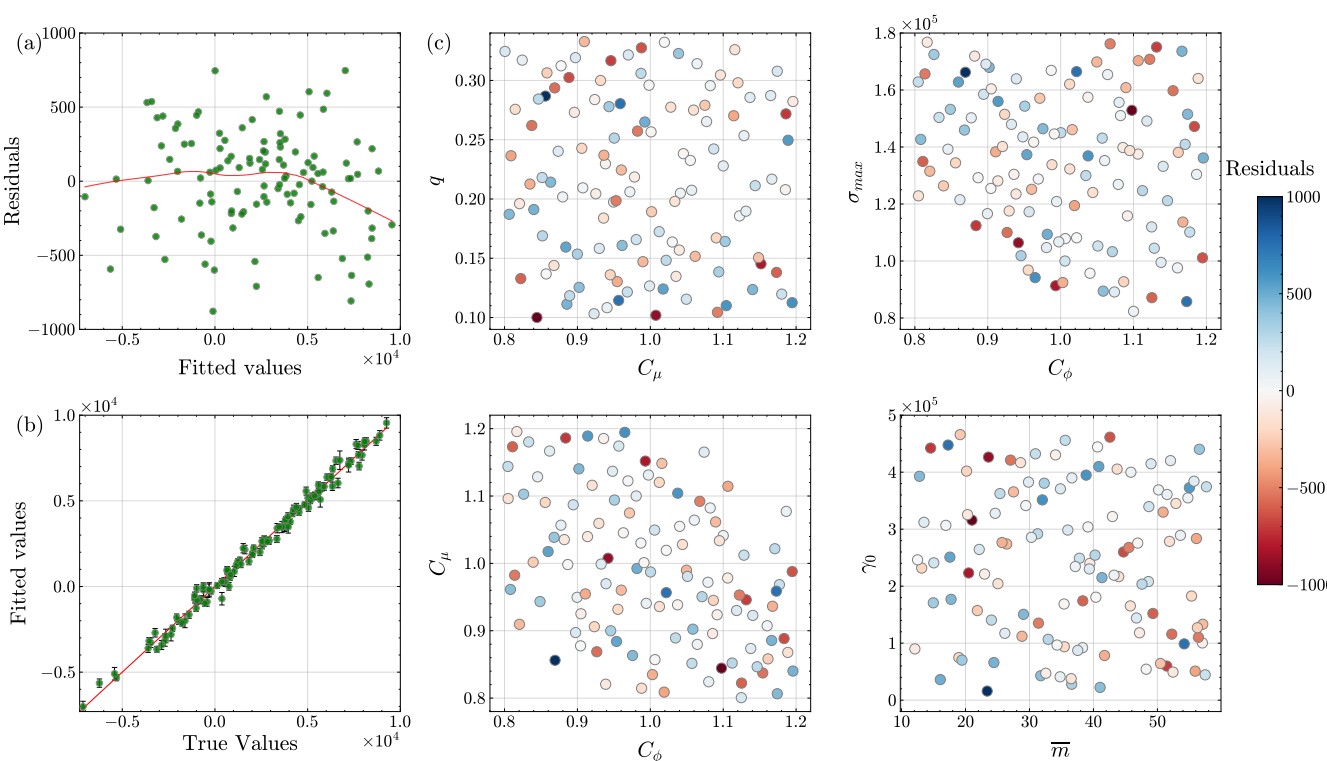

**Figure C5.** Scalar GP emulator fit analysis for volume above flotation change in SSP1 projection ensemble at Year 2300: (a) residuals versus fitted values plot, (b) predicted versus actual values plot, and (c) residuals versus pairwise input parameters plots.





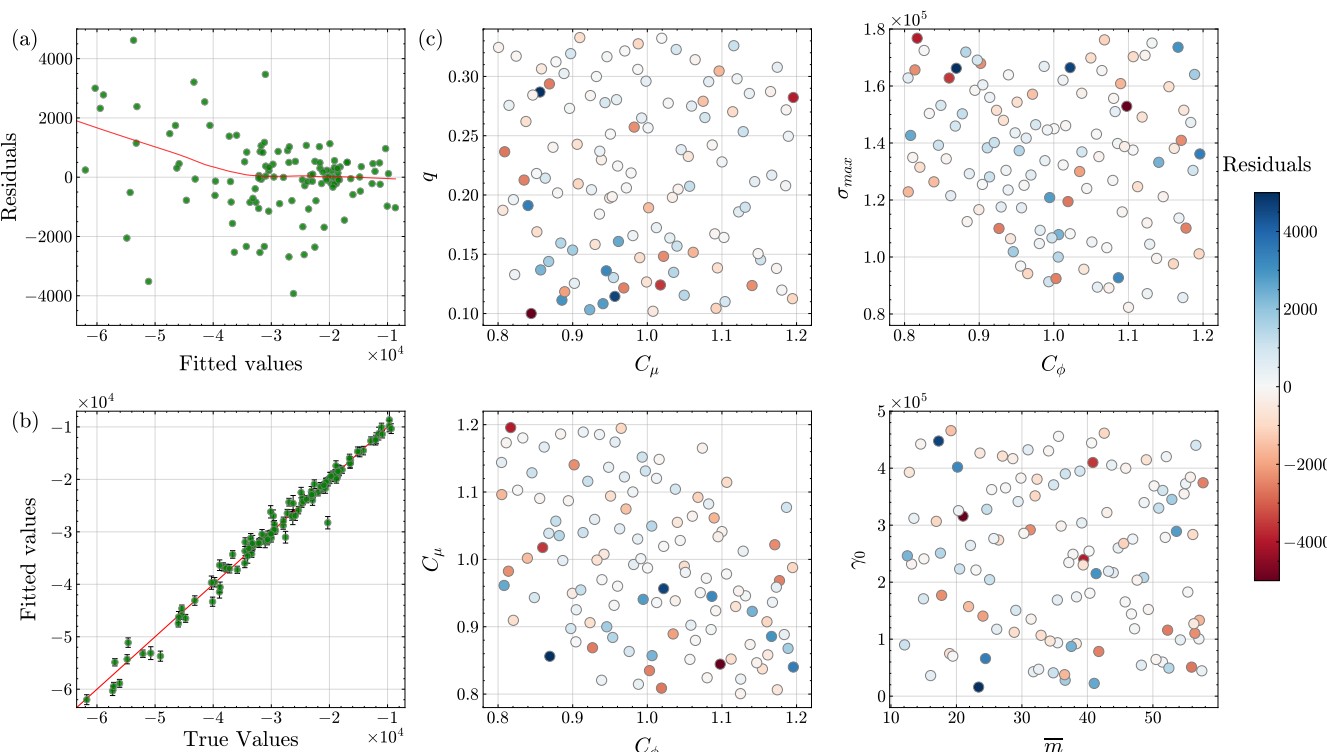

**Figure C6.** Scalar GP emulator fit analysis for volume above flotation change in SSP5 projection ensemble at Year 2300: (a) residuals versus fitted values plot, (b) predicted versus actual values plot, and (c) residuals versus pairwise input parameters plots.

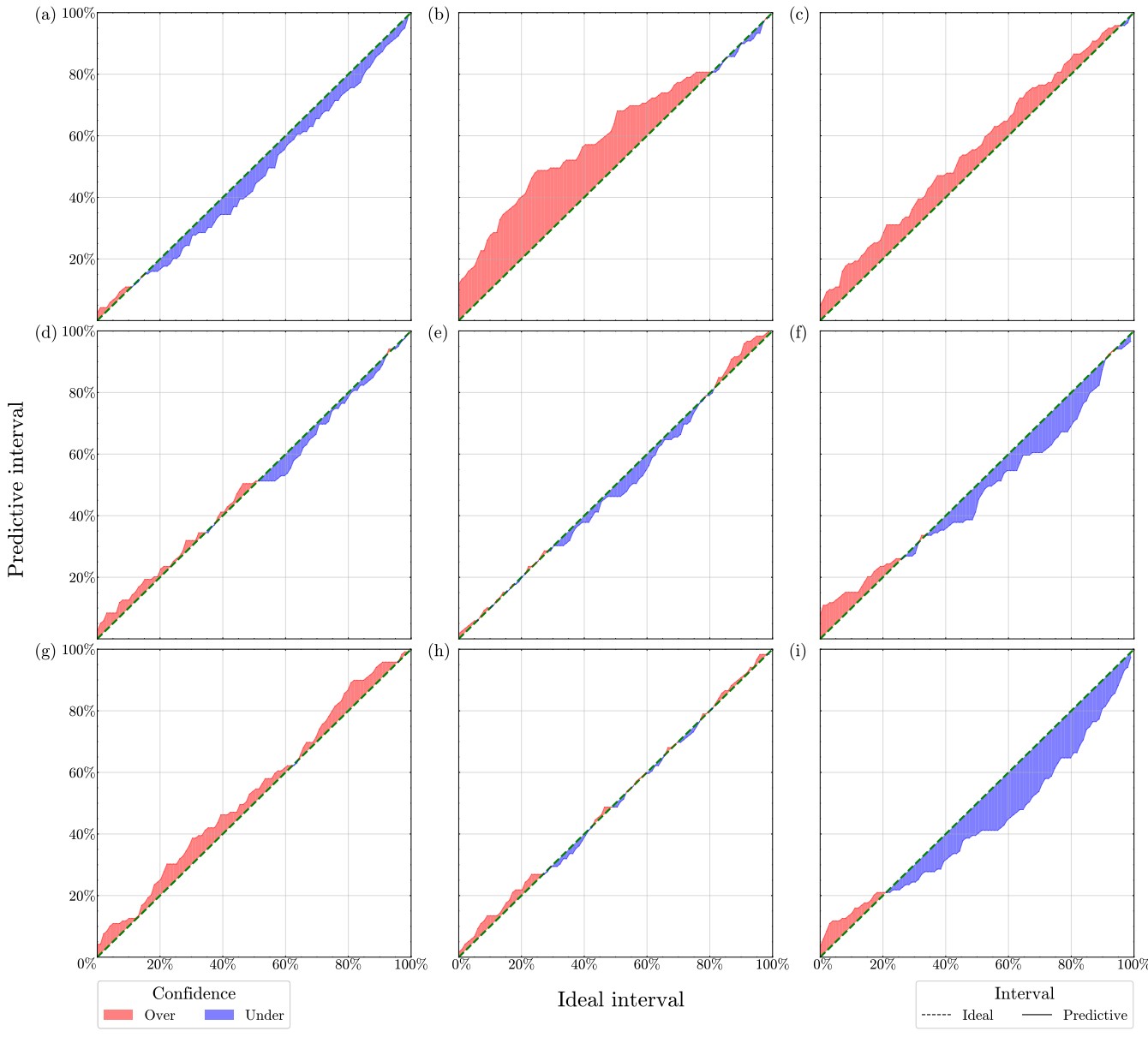

**Figure C7.** Gaussian process emulator predictive coverage results for: (a) calving front position, (b) grounding line position, (c) mass change, (d) PCA emulator in CTRL ensemble, (e) PCA emulator in SSP1 ensemble, (f) PCA emulator in SSP5 ensemble, (g) scalar emulator in CTRL ensemble, (h) scalar emulator in SSP1 ensemble, and (i) scalar emulator in SSP5 ensemble, respectively. Ideal interval denoted by dashed green-line in each panel represents the case of perfect confidence. Overconfidence is represented by red shaded region while underconfidence is represented by blue shaded region in each panel. In panels (a)-(c), emulators are applied in RELX ensemble. In panels (d)-(i), emulators are for volume above flotation change at Year 2300.



## Appendix D: Principal component analysis versus scalar Gaussian process emulator prediction comparison

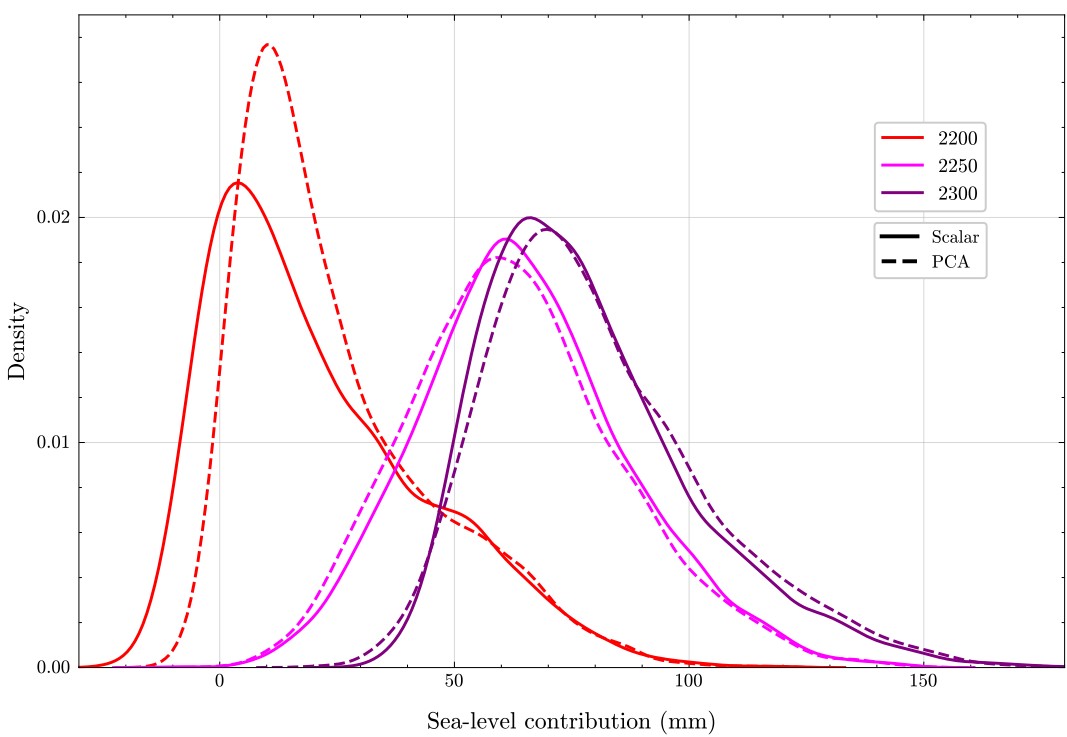

**Figure D1.** Comparison between posterior predictive distributions of future sea-level contribution for SSP5 scenario in Years 2200, 2250, and 2300 obtained using principal component analysis Gaussian process (GP) emulator versus scalar GP emulators fitted to individual year data of the VAF change projections.



*Code and data availability.* Code and data are available on Zenodo at https://doi.org/10.5281/zenodo.11166628.

*Author contributions.* MH, TH, SP, and MP designed the experiments with input and advice from SJ, JDJ, and NMU. TH and MH staged and ran the experiments. SJ ran all the statistical analyses with input from NMU and JDJ. SJ and MH prepared the manuscript with contributions from all the co-authors.

*Competing interests.* The authors declare that they have no competing interests.

*Disclaimer.* This paper describes objective technical results and analysis. Any subjective views or opinions that might be expressed in the
paper do not necessarily represent the views of the U.S. Department of Energy or the United States Government.

*Acknowledgements.* Support for Sanket Jantre, Matthew J. Hoffman, Nathan M. Urban, Trevor Hillebrand, Mauro Perego, Stephen Price, and John D. Jakeman was provided through the Scientific Discovery through Advanced Computing (SciDAC) program funded by the U.S. Department of Energy (DOE), Office of Science, Advanced Scientific Computing Research and Biological and Environmental Research Programs. Simulations were performed on machines at the National Energy Research Scientific Computing Center, a DOE Office of Science
User Facility located at Lawrence Berkeley National Laboratory, operated under Contract No. DE-AC02-05CH11231 using NERSC awards ERCAP0023782 and ERCAP0024081. Brookhaven National Laboratory is supported by DOE's Office of Science under Contract No. DE-SC0012704. Los Alamos National Laboratory is operated by Triad National Security, LLC, for the National Nuclear Security Administration of the U.S. Department of Energy under Contract No. 89233218NCA000001. Sandia National Laboratories is a multimission laboratory managed and operated by National Technology and Engineering Solutions of Sandia, LLC., a wholly owned subsidiary of Honeywell Inter-
national, Inc., for DOE's National Nuclear Security Administration under contract DE-NA-0003525. The authors thank Jeremy Bassis for discussions about calibration of modeled calving.



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
