# Peer review of "Probabilistic projections of the Amery Ice Shelf catchment, Antarctica, under high ice-shelf basal melt conditions"

_EGUsphere, 2024_

## Author Response (AR2)

**Response Letter**

Please find the submitted revised manuscript entitled "Probabilistic projections of the Amery Ice Shelf catchment, Antarctica, under high ice-shelf basal melt conditions," co-authored by Matthew J. Hoffman, Nathan M. Urban, Trevor Hillebrand, Mauro Perego, Stephen Price, John D. Jakeman, and myself.

We would like to thank the editor and two reviewers for their insightful comments and suggestions. The revised manuscript takes into consideration the comments from both the reviewers. A list of significant changes is as follows.

1. As pointed by reviewers 1 and 2, we have provided comments on how the number of ensemble members is determined in our study with a reference on lines 211-214 and 266-267. Further we have included a few statements addressing reviewer 2's concerns about ensemble filtering on lines 306-309.

2. As suggested by reviewer 1, we have now included comments on the choice of Gaussian process kernel and its implications on lines 289-293.

3. As pointed out by reviewer 1, we have provided a justification of the independence assumption among the three observed variables as a remark on lines 324-326.

4. As suggested by reviewer 2, we have included the sensitivity analysis of individual MALI parameters using Sobol' and Shapley sensitivity analysis methods. We have included new subsection 3.6 describing the aforementioned two methods and subsection 4.3 discussing their corresponding results (including Fig. 11 and 12).

5. As suggested by reviewer 2, we now have increased the font sizes of the axis labels, axis tick labels, and legend labels improving the legibility of the figures.

6. As pointed out by reviewer 2, we have added a few statements discussing Fig. 8(c) on lines 426-428.

7. As suggested by reviewer 2, we have modified the colorbar labels of Fig. 1 and 13 (this was the Fig.11 in previous version of the manuscript) improving their legibility.

In the following, we respond to each of the reviewers' comments in detail. The reviewer summary and comments are listed in italic, while our responses are highlighted in blue and not italicized. To aid further review, we have marked all new additions/revisions to our previous manuscript in blue.

**Response to Comments of Reviewer 1**

**Review summary**: *The manuscript discusses the calibration of the MPAS-Albany Land Ice (MALI) model to reduce parametric uncertainties and generate more constrained future projections. Based on a perturbed physics ensemble with 200 runs, the authors have built a GP emulator and used it for Bayesian calibration following the framework of Kennedy and O'Hagan (2001, hereafter referred to as KOH). To propagate the quantified parametric uncertainty in future projections, another set of emulators is used: a PCA-based emulator for the entire trajectory and scalar emulators for certain future time points. The results show that, under the high emission scenario (SSP5), the Amery sector may significantly contribute to sea level rise after the year 2100.*

*Overall, the statistical approaches used for emulation and calibration are well-designed, and the scientific results are an important contribution to the literature on the future of Antarctica. Therefore, the manuscript is suitable for publication in The Cryosphere. I have only a few minor comments:*

1. *200 ensemble members seem to be quite small given that the number of parameters being calibrated is six. Some comments on how the number of ensemble members was determined would be useful.*

   The well-cited paper [1] provides guidelines for determining sample sizes in computer experiments, recommending that, as a rule of thumb, at least 10 data points per input dimension should be used when building an emulator. Given that we have 6 input parameters, this suggests that a minimum of 60 ensemble members would be necessary to build our Gaussian process emulators. Accordingly, in case some had to be discarded, we chose 200 ensemble members to assure there were at least 10 members per input dimension. When executing our study, we discarded 81 runs due to two reasons mentioned in the manuscript: "First, filtering eliminates outliers in potentially complex regions of parameter space that may have reduced the skill of the emulators but would be negligibly sampled based on their low likelihood of matching observations. Second, because in some cases our prior parameter distributions include regions of parameter space that will be negligibly sampled, eliminating runs from these regions reduces the computational cost of the three MALI projection ensembles." This filtering provided 119 ensemble members for RELX Gaussian process (GP) emulator fitting. However, despite discarding a large number of runs, the cross-validation results depicted in Figures 3, 4, and 5 demonstrate that our GP emulators fit the simulation data for each of the three observables quite well. This is supported by the observation that the predictions are in agreement with actual values, and residuals are randomly scattered in each case. Moreover, the emulators trained with 119 ensemble members were more accurate than those trained with all 200 ensemble members (this comparison is not included in our manuscript for brevity). We have included the aforementioned reference and justification on ensemble size selection in the revised manuscript on lines 211-214.

2. *In Section 3.4, the three observed variables are assumed to be independent when defining the likelihood function. I think some comments or justification is needed on this point.*

   We assumed independence among the three observed variables due to the lack of knowledge about the correlation structure among them. We only had a single observation per each of the three observables during the calibration step. Our current calibration framework (and code) can easily accommodate a scenario where observables are correlated, if such information is available. We have provided a justification for the independence assumption as a remark in our revised manuscript on lines 324-326.

3. *The covariance function for the GP emulator is defined as the Matérn with a smoothness of 2.5, for which I commend the authors for avoiding the common mistake of using the squared exponential function. I think it would be even better if they added a brief statement on the implication of this choice, noting that the resulting GP is twice mean square differentiable and hence highly smooth.*

   We chose the Matérn kernel with a smoothness parameter of 2.5 by searching over different kernels and their hyperparameters. Our final choice of Matérn kernel with smoothness parameter of 2.5 ensures that the GP emulator is twice mean square differentiable, providing a high degree of smoothness while avoiding the over-smoothness often introduced by the squared exponential kernel (which renders the fitted GP infinitely differentiable). We included comments highlighting this implication in our revised manuscript on lines 289-293.

**Response to Comments of Reviewer 2**

**Review summary**: *This study seeks to evaluate the future contribution to sea-level rise of the Amery Ice Shelf catchment, and in particular this study takes a Bayesian approach and accounts for parametric uncertainty of AmIS response under different climate scenarios. The authors use Gaussian process emulators to calibrate ensembles of uncertain parameters and then use a second set of emulators to propagate this uncertainty to future sea-level rise contribution. Ultimately, they find that AmIS has the potential to contribute to sea-level rise significantly. The methodology is sensible and well-justified, the results are of great interest to the glaciology community, and the manuscript is well-written. I have a handful of comments below:*

1. ***Ensemble filtering:*** *The authors seem to have removed RELX ensemble members that lie in parts of the parameter space that the emulators may struggle with. I am curious what these parts of the*

*parameter space are, and whether the removal of these regions of the parameter space may affect estimates of posterior uncertainty? This filtering step also seems to have removed a significant portion of the ensemble members (resulting in only 119 members, if I'm understanding the text correctly), which seems to be low. Is this still an appropriate number of members to conduct the calibration?*

As it also addresses this point, please refer to our response to the first comment made by the first reviewer. The filtering process removes 81 ensemble members with low likelihood, specifically those with observable values falling outside the 4-standard-deviation interval. Consequently, excluding these members when constructing the Gaussian process (GP) emulators should have minimal effect on the posterior distributions of input parameters calibrated using GP emulators. While this filtering reduces the emulator's training set, it enhances the fit by eliminating the identified outlier ensemble members. As mentioned in our response to the first reviewer's initial comment, a general rule of thumb often cited in existing literature states that a GP should be built with at least 10 ensemble members per input parameter. We have $\sim 20$ members per input parameter after filtering. Moreover, to not exceed our computational budget, we had to restrict ourselves to 200 member ensembles (before filtering). Nonetheless, the emulator cross-validation plots (Figures 3, 4, and 5) show a good fit for all three GP emulators of the observables. We have provided comments on how the ensemble size is determined with a reference on lines 211-214 and 266-267. Moreover, we have included few statements addressing the concerns about ensemble filtering on lines 306-309.

2. **Effect of parametric uncertainty:** *the authors study the bulk effect of all parametric uncertainty (listed in Table 1) on projections of glacier behavior. Are the authors able to say anything about the contributions of uncertainty in individual parameters on sea-level rise contribution? Does one parameter contribute more than others? If the existing simulations cannot provide this detail, I don't believe the authors need to include it, as there is already quite a bit in this manuscript and the focus is on quantifying overall uncertainty and SLR contribution, but if the existing runs can provide this, this would be useful detail to include.*

Leveraging the independence of the prior distribution of the model parameters, we performed Sobol' global sensitivity analysis on the joint prior to identify how uncertainty in each parameter contributes to sea-level contribution uncertainty in the Year 2300 using scalar GP emulator of volume above flotation (VAF) change for the SSP5 projection ensemble. The first order and total order Sobol' indices for the six input parameters are presented in Fig. 1(a). We observed that the uncertainties in the basal slip exponent ($q$), basal friction scaling factor ($C_\mu$), ice stiffness scaling factor ($C_\phi$), and ice-shelf melt exponent ($\gamma_0$) have nontrivial contributions to the sea-level contribution uncertainty in descending order. Uncertainties in the calving yield stress ($\sigma_{max}$) and ice-shelf basal melt rate ($\overline{m}$) parameters have negligible contribution to the sea-level contribution uncertainty.

[Figure]

Figure 1: Individual MALI parameter uncertainty contribution to the future sea-level contribution uncertainty in the Year 2300 using scalar GP emulator of VAF change for the SSP5 projection ensemble. (a) Sobol' global sensitivity analysis on the joint prior. We present first-order and total-order Sobol' indices for the MALI parameters. (b) Shapley sensitivity analysis on the joint posterior and prior.

In contrast, Fig. 1(b) presents the sensitivity of the model parameters when using the observationally constrained and correlated posterior. Specifically, we present the Shapley indices for each of the input parameters. Sobol' indices cannot be used with correlated variables. We observed that the uncertainties in $q$, $C_\mu$, $C_\phi$, and $\gamma_0$ affect the sea-level contribution uncertainties significantly in descending order. The remaining two parameters have negligible contribution to sea-level contribution uncertainty. In comparison to the Shapely indices computed using the prior distribution of the parameters (also shown in Fig. 1(b)), we observed Bayesian calibration increases the influence of $q$ while reducing the influence of $C_\phi$ uncertainties compared to their prior uncertainties on sea-level contribution.

The aforementioned analysis led to additional interesting findings. First, the $\sigma_{max}$ parameter that was significantly constrained by the Bayesian calibration does not affect sea-level contribution of the ice sheet. Second, the $q$ parameter, which was not constrained by the Bayesian calibration, affects the sea-level contribution the most. To support these statements, the sea-level contribution with their 68% and 95% credible intervals are plotted in Fig. 2. These intervals were estimated by uniformly varying each input parameter over their range and drawing samples of the remaining input parameters from their conditional joint prior distribution. These samples then were propagated through our scalar GP emulator for VAF change in Year 2300 for the SSP5 ensemble then sampled from the emulator's predictive distribution. We observed that $q$ affects the sea-level contribution the most as higher values lead to lower sea-level contribution with compact uncertainty bounds, while lower values lead to higher sea-level contribution with larger uncertainty bounds. Similar trends are observed for $C_\mu$ parameter. The $C_\phi$ parameter affects the sea-level contribution with higher values leading to lower sea-level contribution. Meanwhile, lower values lead to higher sea-level contribution, but the uncertainty bounds stay similar throughout the parameter range. On the contrary, for the $\gamma_0$ parameter, lower

[Figure]

Figure 2: Effect of individual MALI input parameters on sea-level contribution (mm SLE) with uncertainty bounds in the Year 2300 using scalar GP emulator of VAF change for the SSP5 projection ensemble.

values lead to decreased sea-level contribution, while higher values lead to slightly elevated sea-level contribution with uncertainty bounds slightly rising with increasing values for $\log(\gamma_0)$. Finally, varying the $\sigma_{max}$ and $\overline{m}$ parameters has negligible impact on the sea-level contribution. This analysis further supports the findings gleaned from the Sobol' and Shapley analyses.

We have included the aforementioned sensitivity analysis results using Sobol' and Shapley sensitivity analysis methods in the revised manuscript. Specifically, we have included new subsection 3.6 describing the two methods and subsection 4.3 discussing their corresponding results (including Fig. 11 and 12).

3. *In general, the axes labels on the figures are faint, making it difficult to read. If possible, it would be good to bold the text to make it clearer.*

   We have increased the font size to have the axis labels match the text in the manuscript. Other labels (such as axis tick and legend labels) also have been proportionately increased in size. This has improved the letter thickness so the text is easily visible without zooming beyond actual A4 page size.

4. *Figure 8: I did not quite understand what subplot c added to this figure – is it showing the same information as subplot b?*

   The subplot c in Figure 8 was included to indicate how a random sample from the posterior distribution of the six input parameters leads to future sea-level contribution trajectory until Year 2300 when passed through the PCA GP emulator. The 30 trajectories are shown to highlight that the GP produces trajectories similar to those actually simulated in the SSP5 projection ensemble (depicted in red in Figure C3). We have added few statements discussing the Fig. 8(c) on lines 426-428 in the revised manuscript.

5. *Figure 11: the colorbar labels in subplots b and c are very small and hard to read.*

   We have modified the colorbar labels of Fig. 1 and 13 (this was the Fig.11 in previous version of the manuscript) improving their legibility.

**References**

[1] J. L. Loeppky, J. Sacks, and W. J. Welch. Choosing the sample size of a computer experiment: A practical guide. *Technometrics*, 51(4):366–376, 2009.